# A NEW DOG LEARNS OLD TRICKS:
# RL FINDS CLASSIC OPTIMIZATION ALGORITHMS

**Weiwei Kong**
Georgia Institute of Technology
wkong37@gatech.edu

**Christopher Liaw**
University of British Columbia
cvliaw@cs.ubc.ca

**Aranyak Mehta**
Google
aranyak@google.com

**D. Sivakumar**
Google
siva@google.com

## ABSTRACT

We ask whether reinforcement learning can find theoretically optimal algorithms for online optimization problems, and introduce a novel learning framework in this setting. To answer this question, we introduce a number of key ideas from traditional algorithms and complexity theory. Specifically, we introduce the concept of adversarial distributions (universal and high-entropy training sets), which are distributions that encourage the learner to find algorithms that work well in the worst case. We test our new ideas on the AdWords problem, the online knapsack problem, and the secretary problem. Our results indicate that the models have learned behaviours that are consistent with the optimal algorithms for these problems derived using the online primal-dual framework.

## 1 INTRODUCTION

Machine learning has led to dramatic improvements in our capabilities to solve problems previously considered intractable. Besides the obvious empirical evidence of success, there has also been a strong parallel effort in the theory of ML which aims to explain why, when, and how ML techniques work.

Our goal in this paper is to explore whether machine learning can be used to learn algorithms for classic combinatorial optimization problems. We will define this question more specifically by connecting to three concepts from traditional algorithms and complexity theory.

### 1.1 INPUT LENGTH INDEPENDENCE AND THE CONNECTION TO RL

Firstly, by "algorithm," we mean a *uniform algorithm*, one that works for inputs of all lengths, not just for specific input lengths from which the training data is drawn. Typically, models learned using ML techniques tend to be non-uniform, i.e., depend on input length. Previous approaches to finding uniform models — the Neural Turing machines of Graves et al. (2014) and generally the use of recurrent models (including LSTMs) — all suffer from some drawback, most notably the difficulty of training by back-propagation and gradient descent over long sequences. A particularly clever approach, due to Kaiser and Sutskever (2015), adopts the idea of learning "convolution masks" of *finite size* that, when repeatedly applied solve a problem of interest on inputs of arbitrary length; however, the resulting learning problems appear intractable (since the volume of computation grows at least cubically in their setup for most interesting problems, and stable ways to learn convolutions over such large grids are not well-understood). We expand on this point in Appendix F.

Our first *key insight* is that for numerous combinatorial optimization problems, the *primal-dual framework* offers efficient solutions, and also lends itself to efficient *online* algorithms (see, e.g., Buchbinder et al. (2009)) where the input arrives in small units, one at a time, and the algorithm makes a choice about the input (e.g., which advertiser to give a query impression to, which node to match in a graph, whether to include an item in the knapsack, etc.). In addition, there is usually a clear notion of reward associated with a set of decisions, and the goal is often to optimize the

overall rewards collected. This naturally connects our goal to the field of reinforcement learning, and indeed we formulate our learning problems in the Markov Decision Process (MDP) framework and use tools from deep reinforcement learning using policy gradient and DQN methods. Specifically, for any optimization problem, the MDP state will consist of three parts: global parameters of the problem instance (e.g., Knapsack size), a data structure that we expect (and train) the algorithm to learn to maintain (and whose size can depend on the global parameters), and the current input unit. We will train two agents — $U$ that computes an update to the data structure, and $D$ that makes the decision for each input using the data structure and the current input. The RL environment will then carry out the task of applying the update to the data structure, and present it back to the agents $U$ and $D$ as part of the state for the next input. We establish theoretically (Appendix G) that this simple framework is flexible to capture a wide class of problems, and empirically that the resulting algorithms are quite powerful.

## 1.2 Adversarially chosen input distributions

An important question in both ML and Algorithms is what input instances is the algorithm expected to work on. The ML approach is to use a rich enough training set to capture future inputs distributions. Theoretical computer science (TCS), by contrast, traditionally considers *worst-case analysis*: an algorithm is judged by its performance on the worst possible input (specially crafted by an Adversary to beat the algorithm). This approach leads to theoretically robust guarantees on the algorithm. In *stochastic models* of input, the Adversary is somewhat restricted in order to better capture "real" inputs — including the Random Order and the IID models of input.

Our second *key insight* is to bring this approach of adversarial input sets (not to be confused with the notion of adversarial examples which fool ML models, for example, see Goodfellow et al. (2014)) to the ML domain via two techniques to craft training sets:

**1. Universal Training Set**   A common way to prove lower bounds in the TCS literature is to come up with a distribution over inputs and show that *no algorithm* can perform better than some factor $\alpha \leq 1$ compared to the optimal solution, in expectation. This is a key ingredient in the technique of using Yao's Lemma Yao (1977) to prove a lower bound on the performance of all randomized algorithms. For example, in the Adwords problem, there is a specific input distribution which is hard for all online algorithms (Karp et al. (1990); Mehta et al. (2007)). Intuitively, one might expect that if an algorithm does perform well on the specified input distribution then it must have learned some characteristics of the optimal algorithm. We bring this idea to the ML literature by proposing to incorporate such instances into the training.

**2. High-Entropy Training Set**   In some cases, it may difficult to find a universal training set or the universal training set may admit algorithms which perform well on the training set while performing poorly on all other instances. To alleviate this problem we also propose to incorporate training sets that have high entropy. For example, in the Adwords problem, a randomized greedy algorithm is able to perform quite well on the adversarial instance so we incorporate a distribution which is explicitly bad for greedy. In the secretary problem, we provide inputs which come from many different distributions so that it is difficult for it to learn utilize any characteristics of the distributions.

## 1.3 Decoding the Network to find the Algorithm

Our third contribution is the following intriguing question, connected to the broad area of how to interpret ML models. Specifically, suppose that for a given problem, we do manage to learn a network of constant (fixed) size, which does well over inputs of varying lengths coming from varying distributions. Does this allow us to confidently say that the network has learned the correct algorithm? One observation is that since the network is concise, it has to represent a succinct logic. How does that compare to the optimal *pen-and-paper* algorithm that computer scientists have developed for the problem? We will answer such questions by plotting the input-output characteristics of the network learned for the different problems we consider, and compare them to the expected behavior of the traditional algorithms. It may even be possible to convert the network to an algorithmic form, but we leave such an attempt for future work.

## 1.4 SUMMARY OF RESULTS

We study three optimization problems in this work – the *Adwords Problem* (aka Online Budgeted Allocation), the *Online (0-1) Knapsack Problem*, and the so called *Secretary Problem*[1]. All three problems share the feature that they are all very well-studied online combinatorial problems with some probabilistic features, and importantly, the optimal algorithms for each of them have a concise algorithmic form (e.g., not represented implicitly as the solution of a dynamic program).

For all three problems we use RL to find a "uniform algorithm", i.e., an input-length independent logic. We train the models using universal or high-entropy input distributions and find that the models discover the classic algorithms. To mention the highlights of each section:

- Adwords problem: The model learns to find the Balance strategy (Kalyanasundaram and Pruhs, 2000) for unweighted graphs, and the MSVV strategy (Mehta et al., 2007) for weighted graphs which optimally trades-off between load-balancing and greedy strategies.
- Online Knapsack problem: The model learns to find an optimal threshold on *value per unit size* to use to either accept or reject incoming items.
- Secretary Problem: The model learns the optimal "Wait-then-Pick" algorithm which samples the first $1/e$ fraction of the input stream and then picks the next item which is higher than any seen before. It also finds the optimal time-dependent value-threshold algorithm for i.i.d. input.

Our results suggest that it might be possible to draw a formal connection between the online primal-dual framework and RL, e.g., to prove that the online optimization problems solvable in the primal-dual framework admit efficient algorithms learnable via RL. We leave this as a fascinating open question for future work.

*Remark.* In this paper, we use the standard REINFORCE algorithm for policy gradient, with the Adam optimizer. Our contribution is not in extending RL techniques, but in making the connection to algorithms, and showing how standard RL techniques can in fact find the classic "pen-and-paper" algorithms. Further, we do not optimize for the training set-up or hyperparameters; in particular all our training is done over a single machine and training often takes less than a day or two.

## 1.5 RELATED WORK

We are taking a specific angle at the question of how machine learning solves optimization problems. There is a lot of previous work on the larger question of ML and optimization.

A related previous work is that of Bello et al. (2016) which also studies combinatorial problems, particularly the Traveling Salesman Problem and Knapsack, and also uses policy gradient method for an RL framework to optimize the parameters of a pointer network. (This paper also summarizes previous literature on combinatorial optimization using neural networks.) Our work differs in a few ways, but specifically the goal is not only to solve the problem, but also to *interpret* the learned RL policy network and compare to the known optimal algorithms, both in performance and in structure. Moreover, the work of Bello et al. (2016) learns a recurrent network, which could become prohibitively expensive to train on data sets that are large enough to capture the complexity of TSP or Knapsack. Another closely related paper is (Dai et al., 2017), which uses embeddings and RL to find heuristics to solve classic graph problems on specific distributions. The problems they consider are offline in nature, and the heuristics conform to an incremental (greedy) policy guided by scores generated by the RL agent. Specifically, their goal is to find new heuristics for specific distributions, which is different from the work here, where we ask if RL can discover the classic "worst-case" algorithms. Our work is also different in the same way from other work in the space of combinatorial problems, such as that on TSP and Vehicle routing (Kool and Welling, 2018), as well as to the growing literature on using RL to solve optimization for control problems (see for e.g. Lillicrap et al. (2015), Levine et al. (2016)). We also mention as a loosely related paper by Boutilier and Lu (2016), which uses RL (as a budgeted MDP) the solve Budget Allocation problem, although that problem is different from the Adwords problem we consider, in that the question there is to optimally allocate a single advertiser's budget.

---

[1] This is the somewhat unfortunate prevailing name for this problem of Optimal Stopping — we continue using this name in this paper to avoid technical confusion.

One of the goals in this work is to find a uniform algorithm, i.e., one which is independent of the input length, for which we use the RL approach and focus on online optimization problems. As mentioned earlier, there have been several other nice approaches for this problem, each with different difficulty level in the goals, and obstacles in learning. This includes the Neural Turing machines of Graves et al. (2014) and generally the use of recurrent models (including LSTMs), and the "convolution masks" approach of Kaiser and Sutskever (2015).

Finally, from the algorithms literature, our three problems are very well-studied, especially with Knapsack (e.g., Dantzig (1957)) and Secretary (Dynkin, 1963) being decades old problems. The relatively recent Adwords problem (Mehta et al., 2007) is strongly motivated by online advertising (see, e.g., Mehta et al. (2013)), and for which solutions that merge both the theoretical approach and the ML approach could potentially be of high impact in practice of budgeted ad allocations. Algorithmic work on these problems is cited throughout the next sections.

## 2 ADWORDS: ONLINE MATCHING AND AD ALLOCATION

### 2.1 PROBLEM DEFINITION AND ALGORITHMIC RESULTS

We define the AdWords problem (introduced by Mehta et al. (2007) as a generalization of the online bipartite $b$-matching problem) and the key algorithmic results related to this problem.

**Problem 1** (AdWords problem). *There are $n$ advertisers with budgets $B_1, \ldots, B_n$ and $m$ ad slots. Each ad slot $j$ arrives sequentially along with a vector $(v_{1,j}, \ldots, v_{n,j})$ where $v_{i,j}$ is the value that advertiser $i$ has for ad slots $j$. Once an ad slot arrives, it must be irrevocable allocated to an advertiser or not allocated at all. If ad slot $j$ is allocated to advertiser $i$ then the revenue is increased by $v_{i,j}$ while advertiser $i$'s budget is depleted by $v_{i,j}$. The objective is to maximize the total revenue.*

The online $b$-matching problem is the special case when the values are in $\{0, 1\}$.

**Algorithm MSVV** Let $v_{i,j}$ be the value that advertiser $i$ has for ad slot $j$ and let $s_{i,j}$ be the *fraction* of the advertiser $i$'s budget when ad slot $j$ arrives. Define the "tradeoff" function $\psi(x) = e^{1-x}$. Ad slot $j$ is allocated to an advertiser in $\arg\max_{i \in [n]} v_{i,j}\psi(s_{i,j})$ where ties can be broken arbitrarily.

Mehta et al. (2007) showed that when all the values are small compared to the their respective budgets, MSVV obtains at least a $(1 - 1/e)$-approximation of the optimal revenue. Moreover, this is optimal in the worst case. Let us also remark that MSVV has a particular elegant and intuitive form when $v_{i,j} \in \{0, 1\}$. The algorithm is simply to look at the advertisers with a positive value for the ad slot and allocate to the advertiser who has the most fractional budget remaining (reducing to the BALANCE algorithm of Kalyanasundaram and Pruhs (2000))

RL FORMULATION

Suppose there are $n$ advertiser and $m$ ad slots. We formulate the AdWords problem as an RL problem as follows

*State space*: When ad slot $j$ arrives, the agent sees the state $(v_{1,j}, \ldots, v_{n,j}, s_{1,j}, \ldots, s_{n,j})$ where $v_{i,j}$ is the value that advertiser $i$ has for ad slot $j$ and $s_{i,j}$ is the fractional spend of advertiser $i$.

*Action space*: The agent can choose to either allocate the ad slot to an advertiser or not allocate the ad slot at all.

In order to make the input independent of the number of advertisers, we experiment with another method for encoding the input. We relegate the details to Appendix B.

*Reward*: If the action is to allocate ad slot $j$ to advertiser $i$ and the allocation does not cause advertiser $i$ to exceed his budget then the reward for that action is $v_{i,j}$.

*Transition*: If ad slot $j$ was allocated to advertiser $i$ then the advertiser $i$'s fractional spend is updated accordingly. In either case, we move on the next ad slot $j + 1$.

*Architecture and training*: We use a feedforward neural network with five hidden layers each with 500 neurons and ReLU nonlinearity. We then train the network using the standard REINFORCE algorithm with a simple fixed learning rate of $10^{-4}$ and a batch size of 10. To facilitate training, we

use a bootstrapping approach: we first train the network when the number of ad slots is small, say 100 before training it on a larger stream, say 500.

**Special graphs for AdWords** The AdWords problem benefits from the existence of several classes of special graphs which force many algorithms to perform poorly in the worst case. We relegate the details of these graphs to Appendix A.

RESULTS

**Online bipartite $b$-matching** We train the model by feeding it instances of the special graphs defined in the Appendix A. (In fact, we use a uniform distribution over those graphs.) Having chosen the graph, we also randomly choose whether or not to permute the order of the ad slots. We now describe and analyze the output of the learned model to visualize the policy it has learned.

Figure 1 illustrates the algorithm that is learned by the network when training on the mixture distribution that is described above. It is clear that the network has learned some version of balancing although the exact tradeoffs were not realized by the network. We also provide a comparison of the performance of the learned algorithm and the BALANCE algorithm in Table 1. This can be found in Appendix C.

One other interesting aspect to look at is how the duals of the advertisers evolve under the learned agent and under the optimal algorithm. In Figure 5, we see that the trajectory of the duals can be quite similar.

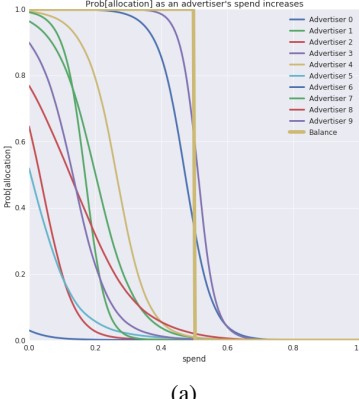
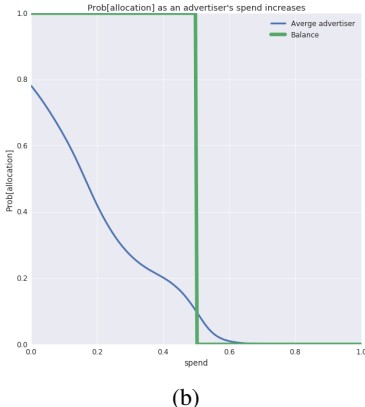

(a)                                         (b)

Figure 1: The algorithm learned by the agent. Each curve in Figure 1a plots the probability that advertiser $i$ (as seen by the network) is allocated as a function of their spend when all other advertiser have spend $0.5$ and all advertiser have value $1$. plots the following curves. Figure 1b is obtained by averaging the curves in Figure 1a.

**Adwords** Finally, we present our results when training our model on AdWords. For training the model on AdWords, we only used the adversarial graph defined in Appendix A. However, for each instance, every advertiser is given a weight $w_i \in (0, 1)$. If the common budget for the advertisers is $B$ then advertiser $i$'s budget is then scaled to $w_i B$ and their value for any ad slot is either $0$ or $w_i$.

Figure 7 in Appendix C plots the policy that is learned by the network. It is clear that the network has learned that, as an advertiser spends more, it also needs to have a larger value before it is allocated the ad slot. Table 4 shows the performance metrics for the learned agent. Note that the agent was only trained on inputs up to length $100$ but it has learned to much larger input lengths. We leave it as future work to find a more adversarial distribution which forces the learner to more accurately recover MSVV.

## 3 Online Knapsack

### 3.1 Problem Definition and Algorithmic Results

**Problem 2** (Online knapsack problem). *Suppose we have knapsack with capacity $B$ and a sequence of $n$ items, represented as a sequence of value-size pairs $\{(v_i, s_i)\}_{i \in [n]}$. The items arrive sequentially and each item must be irrevocably accepted into the knapsack or rejected as soon as it arrives. The objective is to maximize the total value of the items inside the knapsack without violating the capacity constraint.*

**Algorithm "Online Bang-per-Buck"**   When $n$ is large and $\max(v_i, s_i) \ll B$ for all $i \geq 1$, a nearly optimal strategy for the online KP is as follows. For some small $0 < p \ll 1$, accept (when possible) the first $k := \lfloor np \rfloor$ items and define $S(r)$ as the total size of items seen so far with value-by-size ratio (aka "bang-per-buck") at least $r$, i.e. $S(r) = \sum_{i=1}^{k} s_i \mathbb{1}_{\{v_i/s_i \geq r\}}$. Define the threshold ratio $r^* = \arg\min_r \{S(r) < B\}$.

For the remaining items that arrive, accept (when possible) items whose value-to-size ratios are greater than $r^*$. This algorithm is the online version of the natural Bang-per-Buck Greedy strategy for the offline problem Dantzig (1957), and can be interpreted as a "Dual-learning" algorithm, which finds the best online estimate of the corresponding dual variable of the natural linear program.

Finally, as a point of comparison, note that the Knapsack problem is related to the Adwords problem in the following way: it is simpler in that there is only one budget to pack, but it is also harder in that each item has two parameters, the value and the size, while in Adwords one may consider each item to have value to be the same as its size.

### 3.2 RL Formulation

Suppose there are $n$ items arriving in the sequence and the knapsack capacity is $B$. Then an RL formulation that may be used to learn the nearly optimal algorithm from above is as follows. Let $F_s, F_v$ be distributions for the size and values, respectively.

*State space*: At time $i \in [n]$, the agent sees the state $(v_i, s_i, \frac{i}{n}, \frac{S_i}{B})$, where $v_i \leftarrow F_v$ and $s_i \leftarrow F_s$ are the value and size of the $i^{th}$ item, $\frac{i}{n}$ is the fraction of the queue seen so far, $\frac{S_i}{B}$ is the proportion of the knapsack filled by the agent so far. Note that we provide the relative values $\frac{i}{n}$ and $\frac{S_i}{B}$ rather than the absolute values (e.g., of the spend $S_i$) to allow the learning to be scale-invariant.

*Actions*: The agent can choose to either Accept or Reject the item corresponding to the state.

*Transition*: To transition to the next state,

- Draw $(v_{i+1}, s_{i+1})$ from $F_v \times F_s$
- If $S_i + s_i \leq B$ and the action is Accept, then $S_{i+1} \leftarrow S_i + s_i$; else $S_{i+1} \leftarrow S_i$
- Set $i \leftarrow i + 1$

*Reward*: If $S_i + s_i \leq B$ and the action is Accept, then reward is $v_i$; else reward is 0.

*Architecture and Training*: We use a feedforward neural network with 3 hidden layers each with 50 neurons and ReLU nonlinearity. The network is trained using the standard REINFORCE algorithm with a simple fixed learning rate of $10^{-4}$ for the Adam optimizer. The batch size was left at 1.

### 3.3 Results

We train the Policy Gradient RL model on a set of different input parameters. The value and sizes are taken as $(F_s, F_v) \sim \mathcal{U}^2[0, 1]$, and we vary the budget and length of the input sequence to make the KS problem more or less constrained. For each of the input instances, the learned RL policy achieves a performance close to the Bang-per-Buck Algorithm.

We now analyze the output of the learned network to visualize the policy it has learned. Figure 2 plots the probability that an item with a certain value-to-size ratio ($x$-axis) is accepted when it arrives. It is clear that the policy has learned the Bang-per-Buck algorithm with the correct value-by-size threshold for each distribution. For $(B, n) = (20, 200)$ and $(B, n) = (50, 500)$ (Figure 2a,

Figure 2b), there is budget to pick about one-fifth of the items in the stream if we pick at random (recall items have an average size of 0.5), so they have similar thresholds. For $(B, n) = (50, 1000)$ (Figure 2c) the knapsack is much more constrained and can take only a tenth of the items if picked at random, so the network has learned that a much higher threshold is required.

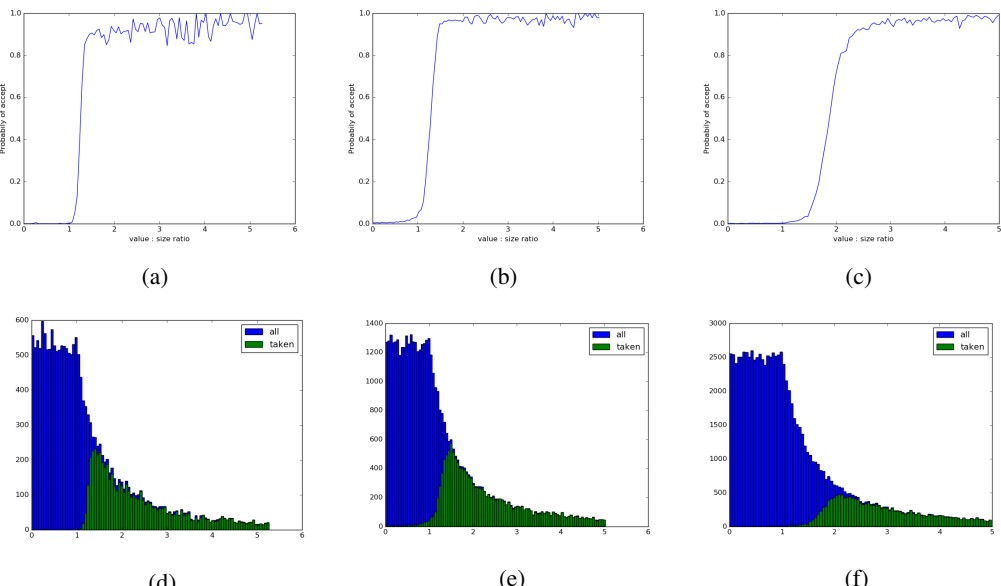

Figure 2: The agent's learned algorithm for the online Knapsack problem where the values and sizes are picked i.i.d. from $\mathcal{U}^2[0, 1]$. The budget and length of sequence vary in the three figures: $(B, n) = (20, 200)$ (left), $(B, n) = (50, 500)$ (center), and $(B, n) = (50, 1000)$ (right). The top row depicts the probability that the agent will accept an item as a function of its value-by-size ratio. The bottom row depicts the histogram of items as a function of their value-by-size ratio ("all" is over all items in the sequence, and "taken" is over only the items that the agent accepts into the knapsack).

### 3.3.1 TOWARDS A UNIVERSAL TRAINING DISTRIBUTION

As opposed to the Adwords and Secretary problems, there is no known theoretical work which provides a universal distribution for the online knapsack problem. However, we do know that a universal algorithm would need to maintain a larger state, for example a histogram of value-by-size ratios of items seen so far, and be able to read the thresholds from the histogram. We take the first steps towards this goal here.

Consider the following distribution: There are two types of knapsack instances, $X$ and $Y$. In both $X$ and $Y$, the budget equals $B$, and all items have value 1. Fix a small positive integer $k$, e.g., $k = 4$. In $X$, items have size either 1 or $k$ with probability $1/2$ each (independently of other items). In $Y$, all items have size either $k$ or $k^2$ with probability $1/2$ each. Finally, the distribution over instances is that we get either an instance of type $X$ or of type $Y$ with probability $1/2$ each.

The point of this distribution is that the optimal solution has a different policy for instances of type $X$ versus $Y$. For $X$ the optimal value-by-size threshold is any number between $1/k$ and 1, while $Y$ the threshold is any number between $1/k^2$ and $1/k$. On the other hand, any single threshold value for the entire distribution will perform sub-optimally for either $X$ or $Y$.

We train our RL agent on this distribution in two different settings:

(A) The original state space defined above, and

(B) The same state augmented by a histogram of the spend binned by the value-by-size ratio. Specifically the state at item $i$ is augmented by $H_i$, an array of length $m$ representing a size-weight $m$-binned histogram of value-to-size ratios seen so far.

The learner (A) without the augmented state does not converge to a good solution, achieving only $73\%$ of optimal, while the learner (B) achieves $95\%$ of the optimal solution quite quickly. A plot of their output (Figure 8 in Appendix D shows that (B) has leveraged the augmented state to be able to determine the optimal threshold for the realized instance type (be it $X$ or $Y$), while (A) has failed to identify which type of instance it got, and uses a single threshold between $1/k^2$ and $1/k$.

We leave for future work the question of leveraging this simple mixed distribution defined above, to find a truly universal training set (e.g., by recursively expanding on it) and show that an RL learner with the augmented state can find a universal bang-per-buck learner (for any input instance distribution).

## 4 THE SECRETARY PROBLEM

### 4.1 PROBLEM DEFINITION AND ALGORITHMIC RESULTS

Problem 3 describes the basic secretary problem.

**Problem 3** (Secretary problem). *There are $n$ candidates with values $v_1, \ldots, v_n$ and an agent that is trying to hire the single best candidate (with the largest value). The candidates arrive in random order and we must irrevocably accept or reject each one before the next one arrives. Once a candidate is accepted, we can not replace by another. The goal is to maximize the probability of selecting the best candidate in the sequence. The algorithm knows the total number of candidates $n$.*

This is an *optimal stopping* problem. We will dispense of the original language and say that *items* arrive according to the above process, and the goal is to pick the item with the largest value.

**The optimal "Wait-then-Pick" Algorithm**  An optimal algorithm for this problem is as follows. First, we reject the first $1/e$ fraction of the items and let $i^*$ be the best amongst these items. Next, we accept the first item $j$ such that $v_j \geq v_{i^*}$. One can show that this algorithm chooses the best item with probability at least $1/e$. It is also known that, with no restriction on the value sequence, no algorithm can do better in the worst case (Dynkin, 1963) (see also Buchbinder et al. (2014)).

We first need to make the input to the models scale-free. We do this by restricting the input values in three different ways, each of them giving a variant of the original problem:

**1. Binary setting**  We start with the original problem. Let $v_1, \ldots, v_n$ be the randomly permuted sequence of numbers. The $i^{th}$ item is presented as a Boolean $m_i$ where $m_i = 1$ if $v_i = \max_{j \leq i} v_j$ and $m_i = 0$ otherwise. That is, $m_i$ represents whether the item has the maximum value among the items seen so far. Note that the Wait-then-Pick algorithm never really cared about the value; only whether a particular value is the maximum value seen in the stream so far. Hence, the Wait-then-Pick algorithm achieves a success probability of $1/e$ and no algorithm can do better.

**2. Percentile setting**  This is a generalization of the binary setting in which item $i$ is represented as a percentile $p_i$ to indicate its *rank* among the items seen so far (so $p_i = 1$, means that the $i^{th}$ item is the maximum so far). Thus this setting provides more information about the stream seen so far. We can show that Wait-then-Pick is still an optimal algorithm achieving a success probability of $1/e$.

**3a. i.i.d. value setting with a fixed distributions**  This is the original setting in which the item values $v_i$ are declared upon arrival, but the restriction is that the values $v_1, \ldots, v_n$ are picked i.i.d. from a fixed distribution $\mathcal{F}$. In this restricted setting, Wait-then-Pick is *not* optimal. Instead, the optimal algorithm is a thresholding algorithm where the threshold decreases over time (Gilbert and Mosteller, 1966, Section 3). Specifically, the algorithm (with knowledge of $\mathcal{F}$ and $n$) determines thresholds $t_1 \geq t_2 \ldots \geq t_n$, and picks the first item $i$ with $v_i > t_i$. This algorithm achieves the optimal success probability.

**3b. i.i.d. value setting with changing distributions**  This is almost identical to the previous setting except that each input instance chooses a distribution $\mathcal{F}$, which may be different every time. The values $v_1, \ldots, v_n$ are drawn i.i.d. from $\mathcal{F}$. Note that the algorithm stated in the previous paragraph no longer works. In particular, forces an algorithm to at least look at some of the input before deciding whether to accept. Thus, this should bring back elements of the Wait-then-Pick algorithm.

RL FORMULATION

In the first three settings, an RL formulation is as follows. At time $i$, the agent sees a **state** $(i/n, x_i)$, where $i/n$ is the fraction of the sequence seen so far, and $x_i = m_i, p_i v_i$ in the binary, percentile, and i.i.d. value setting, respectively. The agent has two possible actions at each state, whether to Accept the item corresponding to the state, or to Reject it. The **transition** at time $i + 1$ for both Actions is simply to pick the next item $(x_{i+1})$ according to the problem setting, and move to $(\frac{i+1}{n}, x_{i+1})$. The **reward** is given only at the end state $(1 + \frac{1}{n}, \Phi)$, where the reward is $+1$ if the agent succeeded in picking the maximum (which is the last 1 in the sequence for the binary case, the last item with percentile 1.0 for the percentile case, and $\max_i v_i$ for the i.i.d. values case) and $-1$ otherwise. Note that our formulation is *not* an MDP as the rewards are not Markovian. Although we can convert it to an MDP with minor modifications to the state, our results show that this is not necessary.

In the value setting with changing distributions, it is impossible to recover the secretary problem with just these two inputs so we augment the state space by providing the maximum value in the past. Otherwise, the RL formulation is as described above.

*Architecture and Training*: We use a feedforward neural network with three hidden layers each with 50 neurons and ReLU nonlinearity. The output layer has two neurons and a softmax is taken over the output logits to obtain the probability of each action. We then train the network using the standard REINFORCE algorithm, with a simple fixed learning rate of $10^{-4}$ and a batch size of 50. However, to facilitate training, we use a bootstrapping approach: We first train the network when the input stream is short, say $n = 10$. Once the learned algorithm is performing sufficiently well, we then increasing $n$, say, by 10, and repeat.

RESULTS

**Binary and Percentile setting** In the binary setting, we trained an agent on instance of secretary up to input lengths of 100. In Figure 3a, we see that the agent has clearly learned a policy which is very similar to the optimal algorithm. In Table 6, we compare the performance metrics of the agent against the optimal secretary algorithm; the learned agent comes quite close. For the percentile setting, Figure 9 again shows that the algorithm has learned to place a sharp threshold. The scores are found in Table 7.

**I.I.D. value setting with a fixed distribution** Recall that in this case, the agent should learn radically different behavior than in the other two settings. Figure 10 shows the learned algorithm for various input lengths and we see that, qualitatively, the agent has learned the optimal algorithm. Here we use the value distribution $\mathcal{U}[0, 1]$. Table 8 compares the optimal and the learned algorithm.

**I.I.D. value setting with changing distributions** In this case, our results show that by using a distribution which has very high entropy (sample $a, b \sim \mathcal{U}[0, 1]$ after which all values are drawn i.i.d. from $\mathcal{U}[\min(a, b), \max(a, b)]$), the model is able to learn a behaviour which is characteristic of Wait-then-Pick, i.e. it waits until some time before accepting any value which is larger than the maximum value seen so far. Somewhat surprisingly, the threshold in our experiments also coincide at $1/e$. This is illustrated in Figure 3b. Table 9 gives he performance metrics. Recall that we augmented the state space so as to provide a "hint" to the learner. We leave it as future work to remove the hint, i.e. the agent should learn to maintain the maximum value it has seen in the past.

## 5    CONCLUSIONS AND FUTURE DIRECTIONS

In this work, we introduced several ideas from traditional algorithmic thinking to train neural networks to solve online optimization problems. In the problems that we consider, our results show that RL was able to find key characteristics of the optimal "pen-and-paper" algorithms. However, in some instances (such as in the knapsack and secretary problem), we saw that some state augmentation was needed in order for the learner to more adequately recover the optimal algorithms. In this work, we took a step towards that by having the RL environment encode that state in a form usable by the agent. In future work, we plan to remove the state augmentation from the RL environment and force the agent to learn the state augmentation as part of the training process.

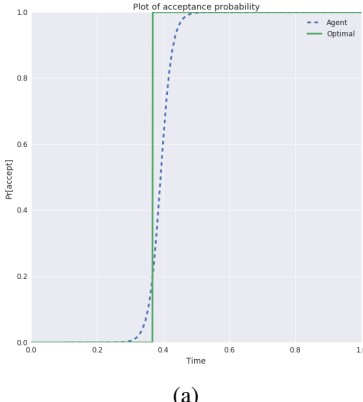
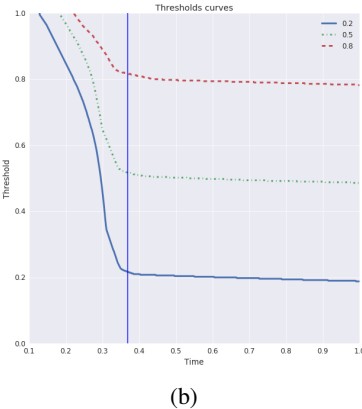

(a)                                                    (b)

Figure 3: Figure 3a compares the agent's learned algorithm with the optimal algorithm in the binary setting. Figure 3b plots the threshold for the agent's learned algorithm in the value setting with changing distributions. Observe that both have learned a threshold at around $1/e$.

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

## A  SPECIAL GRAPHS FOR ADWORDS

Here we describe some special graph for the AdWords problem. Let $U$ denote the set of vertices on the left hand side (corresponding to the advertisers) and $V$ denote the set of vertices on the right hand side (corresponding to the ad slots).

(i) The *adversarial graph* is defined as follows. Let $B$ be an integer and set $B_i = B$ for all $i$. Let $m = Bn$. To define the adversarial graph, we label the ad slots $1, \ldots, m$. For $i \in [n]$, we add an edges between all advertisers in $\{i, i+1, \ldots, n\}$ and all ad slots in $\{(i-1)B + 1, \ldots, iB\}$. Observe that this graph has a perfect $b$-matching by matching advertiser $i$ to ad slots $\{(i-1)B + 1, \ldots, iB\}$. Figure 4a shows an example of the this graph.

It can be shown that for any deterministic algorithm, if one randomly permutes the advertisers then the expected competitive ratio is bounded above by $1 - 1/e$ (Mehta et al., 2007, Theorem 9). Consequently, by an application of Yao's principle (Yao, 1977), for any randomized algorithm, there exists a permutation for which the competitive ratio is bounded above by $1 - 1/e$ (see (Mehta et al., 2007, Theorem 9)).

(ii) The *thick-z* graph is defined as follows. Suppose $n$ is even. Let $B$ be an integer and set $B_i = B$ for all $i$. Let $m = Bn$. Again, label the ad slots $1, \ldots, n$ and the advertisers $1, \ldots, m$. We add edges between advertisers $i$ and $\{(i-1)B + 1, \ldots, iB\}$. Finally, we also add the complete graph bipartite graph between ad slots $\{1, \ldots, Bm/2\}$ and advertisers $\{m/2 + 1, \ldots, m\}$. Figure 4b shows a diagram of this graph

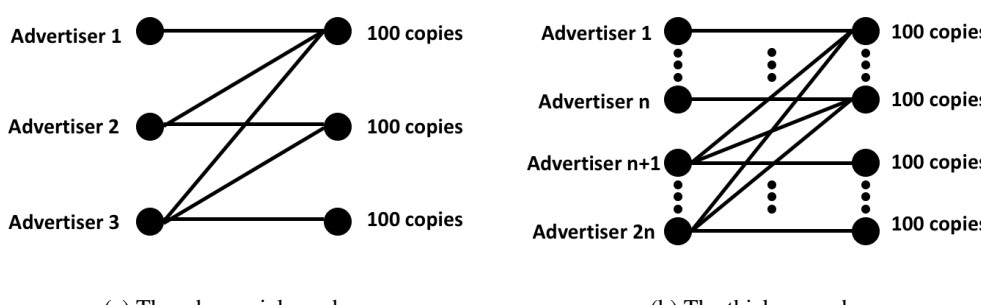

(a) The adversarial graph.  (b) The thick-z graph.

Figure 4: The two special graphs for the online bipartite $b$ matching problem. Figure 4a shows the adversarial graph. In the graph, each advertiser has budget of 100. Hence, there exists a perfect $b$-matching by allocating the first 100 copies to advertiser 1, the second 100 copies to advertiser 2, etc. However, for any randomized algorithm, here is always a permutation of the vertices on the left hand side that will yield a competitive ratio of at most $1 - 1/e$. Figure 4b shows the thick-z graph. Again, each advertiser has a budget of 100 so here exists a perfect matching. However, the greedy algorithm, even if randomized will yield only at most a competitive ratio of $1/2$.

## B RESULTS FOR DISCRETIZED STATE SPACE

For the AdWords experiments, we also considered the following state and action spaces which we dub the *discretized* state and action spaces.

**Discretized, approximate state space:** In order to make our framework applicable to a large number of advertisers, we also introduce a discretized state space. For simplicity, assume the values are in $[0, 1)$. Let $g$ be an integer parameter called the *granularity*. When ad slot $j$ arrives, the agent sees a vector $r \in [0, 1]^{g \times g}$ where for $k_1, k_2 \in [g]$, $r_{k_1, k_2}$ is the fraction of advertisers with value in $[(k_1 - 1)/g, k_1/g)$ and the fraction of their budget spent is in $[(k_2 - 1)/g, k_2/g)$.

**Discretized action space:** The agent chooses $k_1, k_2 \in [g]$. Let $S_{k_1, k_2}$ be the set of advertisers with value in $[(k_1 - 1)/g, k_1/g)$ and the fraction of their budget spent is in $[(k_2 - 1)/g, k_2/g)$. If $S_{k_1, k_2} \neq \emptyset$ then a random advertiser from $S_{k_1, k_2}$ is chosen uniformly at random. The ad slot is then allocated to the chosen advertiser. If $S_{k_1, k_2} = \emptyset$ then the ad slot is not allocated.

### B.1 RESULTS FOR DISCRETIZED STATE SPACE

Figure 6 in Appendix C illustrates the algorithm that is learned by the network. Once again, it is clear that the network has learned to balance so that advertisers who have spent a smaller fraction of their budget are given preference. However we suspect that due to numerical reasons, the network was unable to distinguish between a small fractional number and zero; this is illustrated in Figure 6 where the network did not learn to balance when only most of the bidders are concentrated at spend exactly $0.5$.

Once again, we compare the performance of the learned algorithm and the BALANCE algorithm in Table 2. The table can be found in Appendix C.

## C ADDITIONAL FIGURES AND TABLES FOR ADWORDS

Table 1 compares the performance of the BALANCE algorithm and the learned algorithm when using the basic state space. Note that the learned algorithm was trained with 10 advertisers each with a budget of 50.

Table 2 compares the performance of the BALANCE algorithm and the learned algorithm when using the discretized state space. Note that the learned algorithm was trained with 20 advertisers each with a budget of 20.

In Table 3, we give some experimental evidence that the learned algorithms are uniform in that the quality of the algorithm does not depend too much on the number of advertisers, the budget, or the number of ad slots. Here the agent is trained on input instances with 20 advertisers each with a budget of 20. However, it was tested on instances with varying number of advertisers and varying budgets with up to $10^6$ ad slots. We remark that, due to the discretization, one should not expect to get an approximation of 1 to the BALANCE solution even with the training parameters. Here, we see that the learned agent gets $0.92$ of BALANCE for the training parameters. If an RL learned algorithm is "uniform" then it should not degrade too far below $0.92$ (compared to the BALANCE solution). In our experiments, we see that no matter how long the length of our input is, the quality of its solution never dropped to less than $0.84$, even as we scale up to 1 million ads.

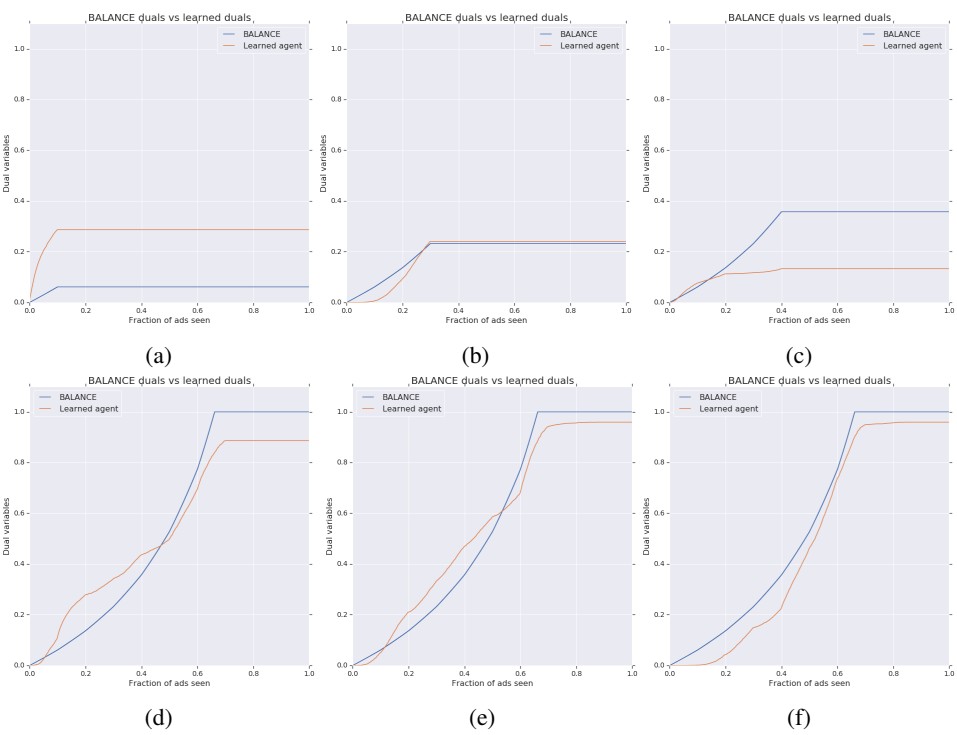

Figure 5: This figure displays the evolution of the advertisers' dual as for BALANCE the learned agent. All the curves correspond to the worst case instance for online bipartite $b$-matching. Figure 5a, Figure 5b, Figure 5c, Figure 5d, Figure 5e, and Figure 5f compares the evolution of the duals for the advertiser that wants the first 10%, 30%, 40%, 70%, 90%, and 100% of the ads, respectively. In many cases, the duals evolve in a similar manner to the optimal algorithm.

Table 1: Comparison of the BALANCE algorithm and the learned algorithm

| Distribution | No. of advertisers | Budgets (common) | BALANCE | Learned |
|---|---|---|---|---|
| Adversarial unpermuted | 10 | 10 | $66.2 \pm 0.1$ | $61.6 \pm 0.3$ |
| | 10 | 20 | $128.4 \pm 0.06$ | $122.6 \pm 1.3$ |
| | 10 | 50 | $326.9 \pm 0.06$ | $300.1 \pm 2.0$ |
| | 10 | 100 | $657.8 \pm 0.06$ | $608.2 \pm 3.4$ |
| Adversarial permuted | 10 | 10 | $90.8 \pm 0.1$ | $77.8 \pm 0.4$ |
| | 10 | 20 | $181.1 \pm 0.2$ | $155.4 \pm 1.8$ |
| | 10 | 50 | $472.3 \pm 0.3$ | $427.7 \pm 2.5$ |
| | 10 | 100 | $962.15 \pm 0.4$ | $770.8 \pm 4.6$ |
| Thick-z unpermuted | 10 | 10 | $70.1 \pm 0.1$ | $67.5 \pm 0.7$ |
| | 10 | 20 | $135.2 \pm 0.05$ | $134.7 \pm 1.5$ |
| | 10 | 50 | $345.5 \pm 0.06$ | $331.5 \pm 1.20$ |
| | 10 | 100 | $695.9 \pm 0.1$ | $663.6 \pm 7.5$ |
| Thickz permuted | 10 | 10 | $91.7 \pm 0.2$ | $82.8 \pm 0.6$ |
| | 10 | 20 | $182.8 \pm 0.3$ | $166.5 \pm 1.2$ |
| | 10 | 50 | $475.6 \pm 0.3$ | $410.1 \pm 2.8$ |
| | 10 | 100 | $867.5 \pm 0.4$ | $832.1 \pm 7.6$ |

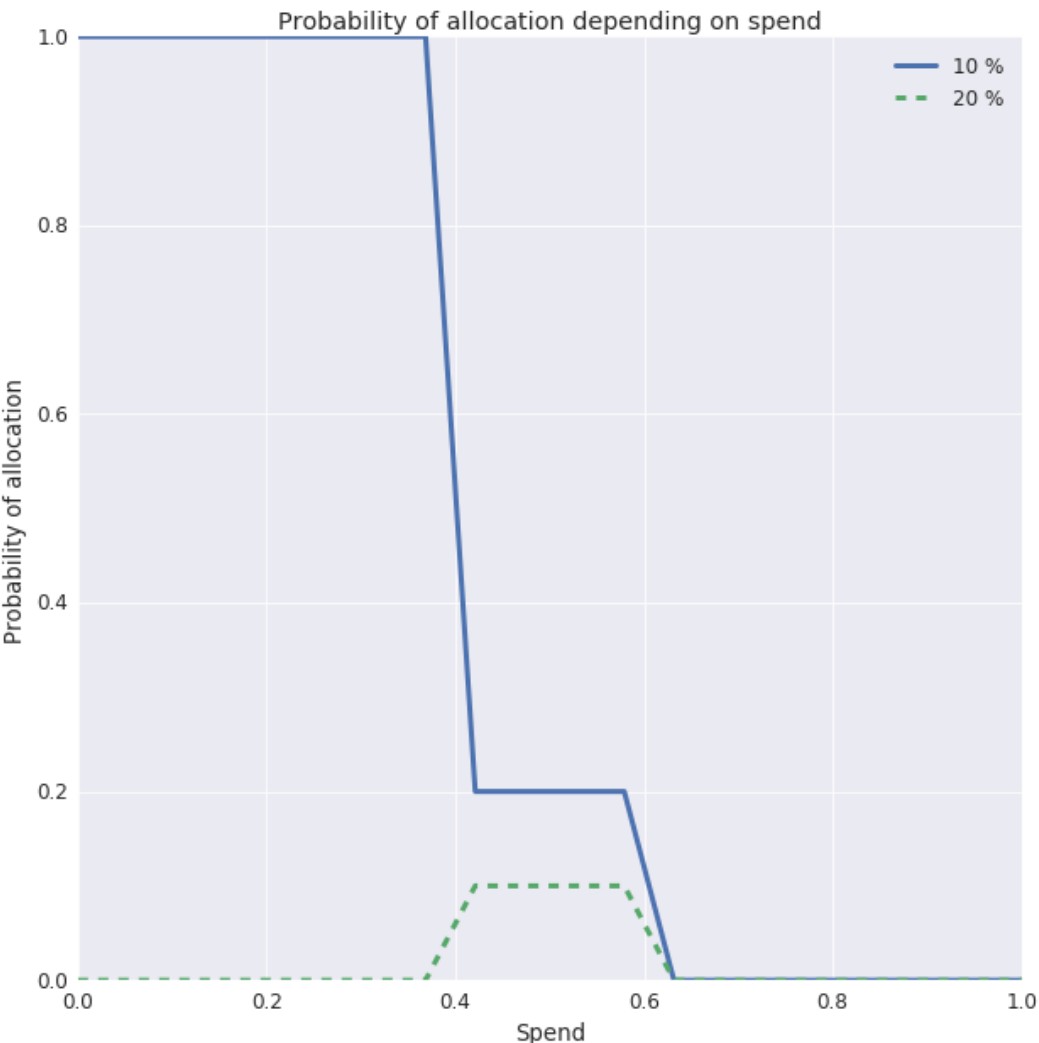

Figure 6: The agent's algorithm learned algorithm for the bipartite $b$ matching problem where the value and spent space has been discretized into 5 buckets. The plots are to be interpreted as follows. In both curves, all advertisers have a value for the ad slot. In the solid curves (resp. the dashed curve) 80% (resp. 90%) of the advertisers have spent exactly 0.5. The plot shows the probability that one of the *other* 20% (resp. 10%) of the advertisers will be allocated the ad slot as a function of their spend. We see that the agent has roughly learned to balance but does have some issues when the number of advertisers in each grid varies substantially.

Table 2: Comparison of the BALANCE algorithm and the learned algorithm with discretized state space.

| Distribution | No. of advertisers | Budgets (common) | BALANCE | Learned |
|---|---|---|---|---|
| Adversarial unpermuted | 10 | 10 | $66.2 \pm 0.1$ | $65.8 \pm 0.2$ |
| | 20 | 20 | $251.2 \pm 0.14$ | $257.2 \pm 0.6$ |
| | 20 | 50 | $640 \pm 0.1$ | $636.7 \pm 1.4$ |
| | 50 | 100 | $1577.2 \pm 2.2$ | $1538.6 \pm 2.2$ |
| Adversarial permuted | 10 | 10 | $90.8 \pm 0.1$ | $84 \pm 0.2$ |
| | 20 | 20 | $361.9 \pm 0.4$ | $329.8 \pm 0.3$ |
| | 20 | 50 | $944.1 \pm 0.4$ | $817.0 \pm 0.5$ |
| | 50 | 100 | $2364.9 \pm 0.5$ | $2026.8 \pm 0.5$ |
| Thick-z unpermuted | 10 | 10 | $70.1 \pm 0.03$ | $66.7 \pm 0.1$ |
| | 20 | 20 | $267.1 \pm 0.1$ | $227.5 \pm 0.4$ |
| | 20 | 50 | $682.7 \pm 0.2$ | $558.8 \pm 3.1$ |
| | 50 | 100 | $2364.9 \pm 0.5$ | $2026.8 \pm 0.5$ |
| Thickz permuted | 10 | 10 | $91.8 \pm 0.1$ | $83.1 \pm 0.2$ |
| | 20 | 20 | $364.3 \pm 0.3$ | $306.1 \pm 0.4$ |
| | 20 | 50 | $949.5 \pm 0.6$ | $758.0 \pm 0.8$ |
| | 50 | 100 | $2370.3 \pm 3.3$ | $1535.7 \pm 1.3$ |

Table 3: This table compares the performance of the learned algorithm compared the BALANCE in the discretized state space. Here, the agent is trained on the adversarial graph with the ad slots arriving in a permuted order. The agent was only trained on the input instance with 20 advertisers and a common budget of 20 but tested on instances with up to $10^6$ ad slots.

| No. of advertisers | Budgets (common) | No. of ad slots | Approx. of BALANCE |
|---|---|---|---|
| 10 | 10 | 100 | 0.9 |
| 20 | 20 | 400 | 0.92 |
| 30 | 30 | 900 | 0.88 |
| 10 | 2000 | 20000 | 0.85 |
| 10 | 4000 | 40000 | 0.85 |
| 25 | 4000 | 100000 | 0.84 |
| 50 | 400 | 20000 | 0.84 |
| 100 | 100 | 10000 | 0.85 |
| 100 | 1000 | 100000 | 0.85 |
| 200 | 100 | 20000 | 0.85 |
| 500 | 50 | 25000 | 0.85 |
| 1000 | 100 | 10000 | 0.84 |
| 25 | 40000 | 1000000 | 0.84 |

Table 4: Comparison of the MSVV algorithm and the learned algorithm with discretized state space.

| Distribution | No. of advertisers | No. of ad slots | MSVV | Learned |
|---|---|---|---|---|
| Adversarial graph, ad slots permuted, values in $\{0, 1\}$ | 5 | 50 | $45.7 \pm 0.1$ | $45.1 \pm 0.1$ |
| | 5 | 100 | $91.0 \pm 0.2$ | $91.4 \pm 0.3$ |
| | 5 | 500 | $482.3 \pm 0.3$ | $462.6 \pm 1.4$ |
| | 5 | 1000 | $976.0 \pm 0.6$ | $930.0 \pm 2.7$ |
| Adversarial graph, ad slots permuted, values drawn randomly | 5 | 50 | $23.1 \pm 0.7$ | $22.4 \pm 0.7$ |
| | 5 | 100 | $45.4 \pm 1.0$ | $46.3 \pm 1.0$ |
| | 5 | 500 | $238.4 \pm 5.7$ | $240.0 \pm 6.1$ |
| | 5 | 1000 | $471.4 \pm 7.5$ | $474.2 \pm 7.4$ |

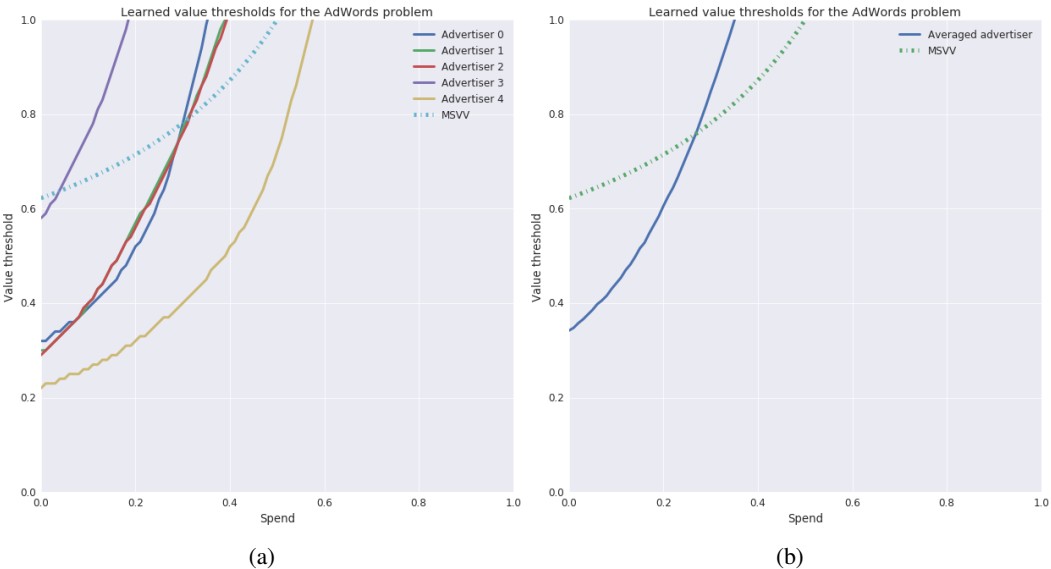

(a)              (b)

Figure 7: The algorithm learned by the agent. Figure 7a plots the following curves. Fix advertiser $i$. Then all advertisers except $i$ has value 1 for the ad slot and their fractional spend is $0.5$. We then let the fractional spend of bidder $i$ vary from $0$ to $1$ and plot the minimum value that advertiser $i$ needs to be allocated the item with probability at least $0.5$. The dotted curve corresponds to the threshold given by MSVV. Figure 7b is obtained by averaging the curves for all the advertisers.

# D    ADDITIONAL FIGURES AND TABLES FOR KNAPSACK

Table 5: Comparison of the Bang-per-Buck knapsack algorithm and the learned algorithm.

| No. of items in sequence | Budget | Bang-per-Buck | Learned | Average Performance |
|---|---|---|---|---|
| 200 | 20 | $51.17 \pm 0.31$ | $49.45 \pm 0.61$ | 96.63% |
| 500 | 50 | $128.61 \pm 0.47$ | $124.75 \pm 1.02$ | 96.94% |
| 1000 | 50 | $182.15 \pm 0.61$ | $174.87 \pm 1.42$ | 96.02% |

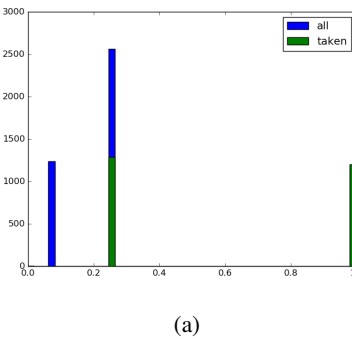

(a)

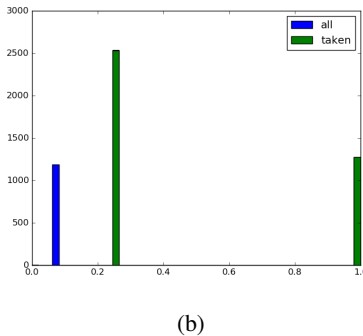

(b)

Figure 8: In Figure 8a, the learner with augmented state accepts only items of size 1 for type $X$, and only items of size $k$ for type $Y$. In Figure 8b, the learner without the augmented state accepts items of size 1 and $k$ for type $X$ (which is suboptimal for $X$), and only of size $k$ for type $Y$ (which is optimal for $Y$).

## E ADDITIONAL FIGURES AND TABLES FOR SECRETARY

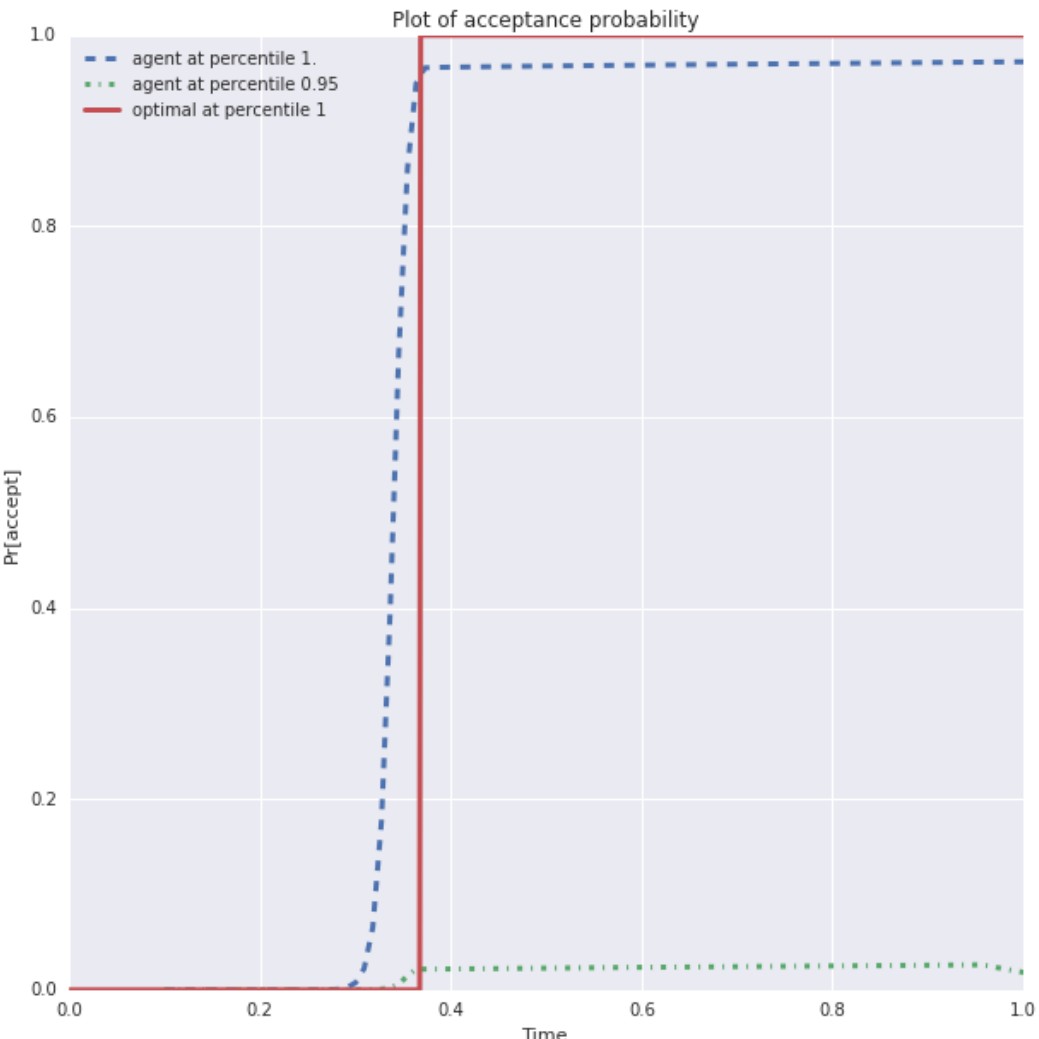

Figure 9: The agent's algorithm for the secretary problem compared with the optimal algorithm for the secretary problem.

Table 6: Comparison of optimal algorithm and learned algorithm (scores are mean $\pm$ standard deviation)

| $n$ | Optimal algorithm | Learned algorithm |
|---|---|---|
| 100 | $0.3733 \pm 0.005$ | $0.371 \pm 0.015$ |
| 2000 | $0.372 \pm 0.014$ | $0.372 \pm 0.014$ |

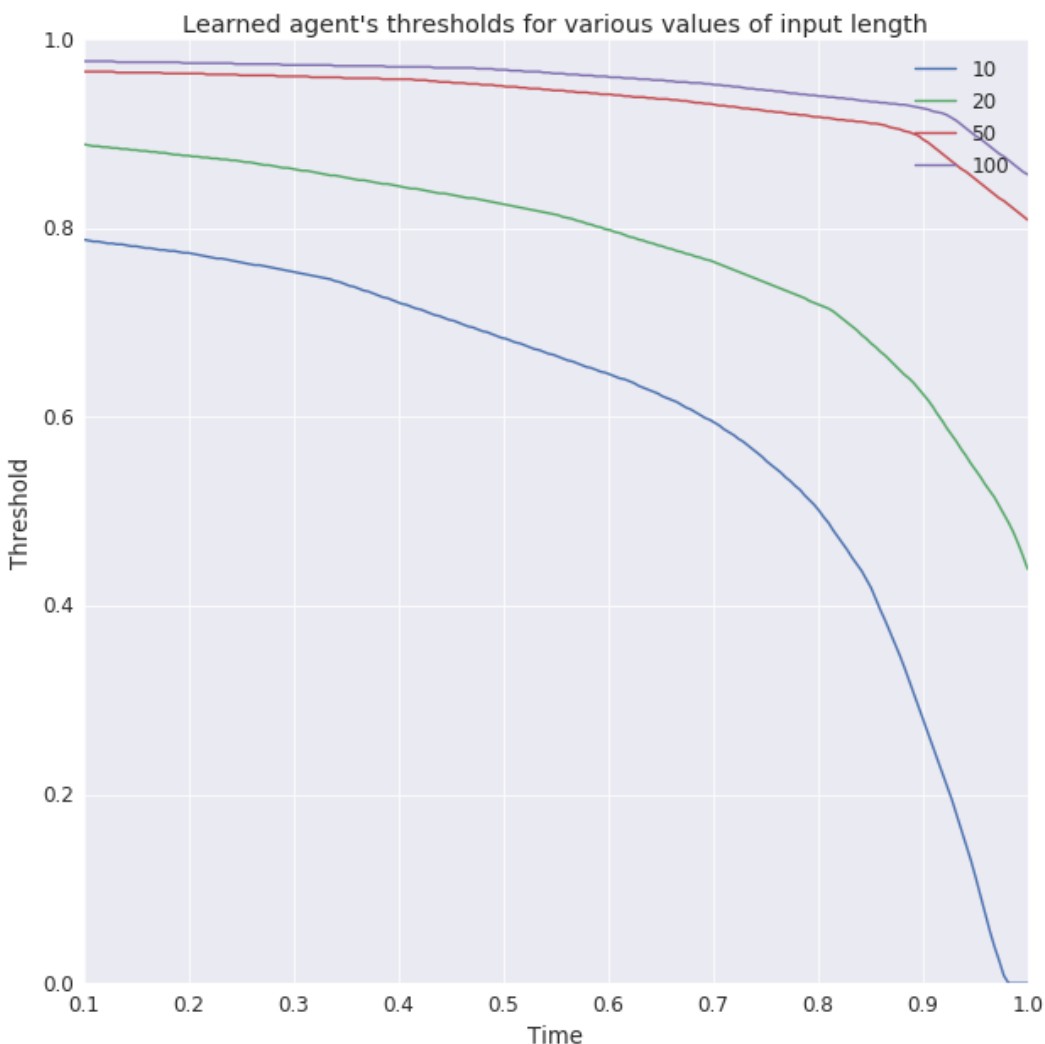

Figure 10: The agent's algorithm for the secretary problem with i.i.d. values for various values of the input length $n$.

Table 7: Comparison of optimal algorithm and learned algorithm (scores are mean $\pm$ standard deviation)

| $n$ | Optimal algorithm | Learned algorithm |
|-----|-------------------|-------------------|
| 10 | $0.393 \pm 0.018$ | $0.396 \pm 0.014$ |
| 20 | $0.376 \pm 0.017$ | $0.386 \pm 0.011$ |
| 50 | $0.371 \pm 0.014$ | $0.371 \pm 0.015$ |
| 100 | $0.370 \pm 0.022$ | $0.358 \pm 0.015$ |

Table 8: Comparison of optimal algorithm and learned algorithm. The scores obtained by the algorithm are taken by averaging 10 runs with each run traversing through 1000 iterations.

| $n$ | Optimal algorithm | Learned algorithm |
|-----|-------------------|-------------------|
| 10 | 0.608699 | $0.584 \pm 0.012$ |
| 20 | 0.5942 | $0.571 \pm 0.015$ |
| 50 | 0.585725 | $0.551 \pm 0.012$ |
| 100 | $\approx 0.58$ | $0.530 \pm 0.014$ |

Table 9: Comparison of optimal algorithm and learned algorithm

| $n$ | Optimal algorithm | Learned algorithm |
|-----|-------------------|-------------------|
| 10  | $0.393 \pm 0.018$ | $0.404 \pm 0.010$ |
| 20  | $0.376 \pm 0.017$ | $0.366 \pm 0.016$ |
| 50  | $0.371 \pm 0.014$ | $0.331 \pm 0.021$ |
| 100 | $0.370 \pm 0.022$ | $0.267 \pm 0.021$ |

## F    Background on Algorithms and Input Length Independence

In this section, we visit a fundamental question that motivates our work at a very high level: *what* is an algorithm? We present a detailed and somewhat informal description of the various nuances that make the question of *learning* an algorithm challenging and subtle.

Traditionally in Computer Science we define an algorithm as a finite piece of code (for some machine model) that gives a recipe to solve all instances of a problem. This definition works perfectly when the underlying model of computation is a Turing machine, a standard RAM machine, or a logical system like first-order logic. In particular, for all these models, the same algorithm works for instances of all possible sizes. By contrast, when the model of computation is non-uniform, e.g., Boolean or arithmetic circuits or branching programs or straight-line programs or feed-forward neural networks, this notion breaks down. In these models, it is customary to think of a concrete computation (algorithm) as operating on inputs of a fixed length.

Typically, the models learnt using various machine learning techniques tend to be non-uniform. Even some of the most basic ML models such as linear or logistic regression use this notion implicitly: given pairs $(x_1, y_1), ..., (x_m, y_m)$, where each $x_i \in \mathbb{R}^n$, a linear model $w = (w_1, ..., w_n) \in \mathbb{R}^n$ that is designed to minimize $\sum_i \|\langle w, x_i \rangle - y_i\|_2^2$ works only for inputs of length $n$ (and aims to work *well* for inputs of length $n$ from a distribution that supplies the training data). Similarly, feed-forward neural networks commonly employed for various image classification tasks work on inputs of fixed dimensions (we will discuss an exception to this momentarily).

Given this state of affairs, what does it mean to learn an algorithm for a problem that is well-defined for inputs of arbitrary length? Moreover, is it even reasonable to expect that inputs trained on bounded length be able to generalize to inputs of arbitrary length? We next discuss a few specific success stories and a few attempts in machine learning that have failed to yield satisfying solutions.

In the pre-machine-learning era, an early success story is that of finite-state machines and regular expressions. It is possible (Hopcroft and Ullman, 1979), in principle, to learn an FSM (equivalently, a regular expression) if we are given the correct label for all instances of (a finite) length bound (a bound that depends only on the language). Even for the next rung on the Chomsky hierarchy, namely context-free languages, the situation is extremely murky (see Hopcroft and Ullman (1979)), and depends delicately on the type of training examples, the structure of the grammar, etc. (The fundamental question of whether two given context-free grammars are equivalent is undecidable, and this type of intractability is closely associated with the problem of inferring or learning grammars from labeled data.) The situation is entirely hopeless for Turing machines, and quickly runs into issues of undecidability.

In the context of neural networks (or equivalently differentiable arithmetic straight-line programs), three developments are worth highlighting:

1. The Neural Turing Machine model of Graves et al. (2014) offers a philosophically complete answer to the question of what it means to learn algorithms. The model is fundamentally a recurrent neural network with a finite number of parameters, and is Turing-complete in the parlance of artificial intelligence, that is, it is as powerful as the standard Turing machine model. While the work of Graves et al. (2014) has many impressive examples of what these models can be trained for (for example, by training a model to copy short sequences of numbers, it has learned to copy longer sequences of numbers with relatively small error), they are quite far from being trainable for significantly more complex algorithmic tasks. A fundamental bottleneck here is that recurrent networks, in general, are very hard to train reliably through back-propagation over long input sequences. This problem exists even with cleverly crafted variants of recurrent neural networks like LSTMs (Hochreiter and Schmidhuber, 1997) that have been successful in practice in dealing with sequences of hundreds of input symbols.

2. The idea of convolution that is commonly used in image-processing tasks (including feed-forward neural networks for various image-related tasks such as classification, object identification, etc.) offers, in principle, a method to define finite-size algorithms for inputs of arbitrary length. A convolution mask is a (short) sequence of finite size that is applied to every contiguous patch of the input sequence, emitting a finite-size sequence of symbols each time. This results in possibly increasing the input size, but in practice it has been observed that the following paradigm works very well in practice (especially for image-related problems): perform several (but fixed number of) layers of

convolution then pool the resulting sequence into a fixed-length summary, and finally apply an expensive neural network computation on this fixed-length summary to produce the output. The key here is that the pooling operator is typically defined as dividing the input into a fixed number of regions (possibly overlapping) and applying a simple differentiable function (e.g., the SUM or addition operator) to the convolution outputs in each region. In particular, the architecture defined above implies that regardless of the size of the input instance, the goal of learning is to infer a fixed number of parameters, and equally importantly, the depth of the resulting computation graph is finite, so algorithms like back-propagation have a chance to succeed.

Unfortunately, however, the finite-depth limitation that enables (potentially) efficient (or at least feasible) learning, comes with a severe cost: it is unclear how rich the resulting model is, that is, we dont know if there are algorithms for interesting tasks in this model. This question is related to fundamental questions in computational complexity theory: on the one hand, the closest complexity class that captures computations like this, namely $TC^0$ (the class of problems solvable by constant-depth polynomial-size circuits with AND, OR, NOT, and THRESHOLD gates), is not known to be powerful enough to perform all polynomial-time computations (or even logspace computations) (Aaronson); on the other hand, a slight weakening, where we drop THRESHOLD gates, results in the complexity class $AC^0$ that is not powerful enough to compute even simple functions like the parity of n bits (or the product of two n-bit integers). To summarize, we dont know if the convolution-pooling paradigm(even though it works well in practice on certain classes of problems) is powerful enough to represent nontrivial algorithms for interesting real-world computational tasks.

3. The work of Kaiser and Sutskever (2015) attempts to go beyond this type of limitation using an elegant idea: they go back to first principles in terms of how Turing machines work, and propose a model that captures it well. In their model, the goal is to learn a finite-sized set of convolution masks that, when repeatedly applied to an input (so its a recurrent network), effectively solves the problem. In other words, they remove the depth limitation in the convolution-pooling model outlined above. This immediately restores the power of the model; it is now rich enough to simulate a Turing machine with polynomial overhead. On the other hand, even simple tasks like adding two n-bit integers could now result in $\Omega(n^2)$ or $\Omega(n^3)$ in terms of the volume of the computation (volume refers to the product of time and space). The resulting depth makes this hard to train, but at least this solves the non-uniformity problem: in principle, one can train a model on inputs of length 100 and hope that it will work on arbitrarily long inputs. Kaiser and Sutskever present some impressive examples of basic problems for which they are able to learn algorithms (e.g., addition, multiplication, etc.). In our view, this work comes closest to addressing the philosophical questions in the right framework, even if the resulting learning problems appear intractable.

# G  SYMMETRIC FUNCTIONS AND LEARNING UNIFORM ALGORITHMS

One of our goals in this paper is to understand under what conditions we can learn *uniform* algorithms for various problems. By *uniform* algorithms, we mean algorithms that work for inputs of all lengths. In particular, this implies that the number of parameters that describe the algorithm (that we wish to learn) needs to be *finite*. In this section, we show that uniform algorithms can be learned for a large and interesting class of problems by combining two key ideas.

(1) As described in the Introduction, we focus on learning algorithms for optimization problems such as the classic Knapsack problem, matching and allocation problems in bipartite graphs, and versions of the "secretary problem". By focusing on optimization problems which have a clear notion of immediate and/or overall rewards, we are able to cast the learning problem as a reinforcement learning (RL) problem. This helps us effectively sidestep the "depth problem" which arise from training recurrent networks.

(2) We focus on the class of functions which can be computed, or approximated by, algorithms utilizing a small amount of memory and only a few passes over the input sequence (for example, online algorithms with space constraints).

Although these two restrictions may be limiting, they already capture some interesting algorithmic problems that admit nontrivial solutions. For instance, the AdWords problem has an optimal algorithm that looks only at the current state of nature and ignores all past information; and the online knapsack problem has a nearly optimal solution which requires only $O(1)$ memory. Importantly, both of these problems require memory which is *independent* of the input length.

These two restrictions are certainly limiting, but as we shall see shortly, they lead to a very interesting sweet spot that includes numerous real-world optimization problems. For instance, the AdWords problem has an optimal algorithm that looks only at the current state of nature and ignores all past information; and the online knapsack problem has a nearly optimal solution which requires only $O(1)$ memory. Importantly, both of these problems require memory which is *independent* of the input length. Equally importantly, they lead to us a representation theorem that establishes rigorously that such problems can be solved by computation graphs of constant depth (independent of input length), which, in turn, leads to tractable learning tasks.

A few remarks are in order on the memory-restricted computational models we focus on. The simplest model is the standard "streaming algorithm (Alon et al., 1999; Henzinger et al., 1998), where the algorithm is endowed with a small amount of working memory (typically polylogarithmic in the input length) and processes the input sequence one element at a time, spending very little time on each input item (typically polylogarithmic in the input length). This model has been known in the CS theory literature to be surprisingly powerful in estimating several statistical quantities of input streams (see Alon et al. (1999)). While powerful, this model is somewhat cumbersome when one wishes to solve problems whose output is more than a handful of numbers, e.g., the matching problem in bipartite graphs. The prevailing model of choice for such problems, originally proposed by Muthukrishnan (2005), is the semi-streaming model. In this model, the algorithm is still a one-pass (or few-pass) algorithm, but the algorithm is allowed linear amount of memory (linear in the right complexity measure, e.g., number of vertices for graphs) but still needs to process each input item in polylogarithmic time. A particular variant of this model that we will discuss is what well call segmented semi-streaming model where the linear amount of memory is further assumed to be divided into constant-sized units, one per variable of interest (e.g., a handful of variables per vertex of a graph). This is a very natural model of computation that is also quite rich in what can be accomplished in it: for example, there are powerful algorithms for various versions of matching and allocation problems (Karp et al., 1990; Mehta et al., 2007; Agrawal et al., 2018) (which are some of the "hardest" problems in P, the class of polynomial-time solvable problems). Informally this makes sense – many algorithms maintain per-variable information throughout the course of their execution (weights, dual variables, vertex color, etc.).

In ML terms, we may informally think of a streaming algorithm as the discrete analogue of an LSTM. This, of course, makes a streaming algorithm (never mind the more powerful variants like semi-streaming) difficult to learn via back-propagation; the key twist, however, is a structural representation theorem (established in the CS theory literature (Bar-Yossef et al., 2002; Feldman et al., 2010) and independently re-discovered in the context of deep learning by Zaheer et al. (2017)) which shows effectively that every symmetric function computable in the streaming model is also computable in the weaker "sketching model (where each input item is sketched independently of each other, and the resulting sketches are aggregated by a combining function). Informally, this gives us a large class of functions that can be computed by computation graphs of constant depth with a constant number of parameters that need to be learnt. This is the key fact that makes our framework tick.

**Definition 1.** A function $f$ is said to be computable by a *streaming algorithm with space* $s(\cdot)$ if there is an algorithm $A$ that for every $n$ and every $x = x_1, \ldots, x_n$ computes $f(x)$ given *one-way* access to $x$ (that is, reads $x$ one coordinate or "item" at a time), uses space no more than $s(n)$.

(Traditionally, we also require that $A$ runs in time $\mathrm{poly}(s(n))$ on each item of the input stream; for the purposes of this paper, this is unimportant, so we will assume that each input item has constant size and $A$ processes each item in constant time.)

**Definition 2.** A function $f$ is said to be computable by a *sketching algorithm* if there are two (uniform) algorithms $S$ and $R$ such that for every $n$ and every $x = x_1, \ldots, x_n$, $f(x_1, \ldots, x_n) = R(S(x_1), \ldots, S(x_n))$. Here $S$ is called the "sketching function" and $R$ is called the "reducer function".

A function that is computable by a sketching function is thus computable in a simple "Map-Reduce" style of computation (Dean and Ghemawat, 2008).

The main idea of this section is that while differentiable streaming algorithms are hard to learn (because on long inputs, the back-propagation chains are too long), differentiable sketching algorithms are easy – there is a finite number of parameters to learn, no back-propagation chain is more than a

constant number of steps long, and we can train networks that do the "sketching" (like the function $S$ in the definition above) and the "reducing" (like the function $R$ above) on inputs of arbitrary length, provided $R$ is simple enough. This leads

The key complexity-theoretic result we draw on is the following theorem (Bar-Yossef et al., 2002; Feldman et al., 2010), which shows that under suitable conditions, any function computable by streaming algorithms are also computable by sketching algorithms. Essentially, this result has also been independently discovered in the "deep sets" work of Zaheer et al. (2017).

**Definition 3.** A function $f$ on $n$ inputs $x_1, \ldots, x_n$ is said to be *symmetric* if $f$ is invariant to permutations of the inputs, that is, for all $n$, all $x = x_1, \ldots, x_n$ and permutations $\pi \in S_n$ (the group of permutations of $n$ objects), $f(x_1, \ldots, x_n) = f(x_{\pi(1)}, \ldots, x_{\pi(n)})$.

**Theorem 1** (Bar-Yossef et al. (2002); Feldman et al. (2010) paraphrased)**.** *If a function $f$ is symmetric and is computable by a streaming algorithm, it is also computable by a sketching algorithm.*

There are additional technical constrains in the results of Bar-Yossef et al. (2002); Feldman et al. (2010); Zaheer et al. (2017), but from the viewpoint of learning uniform algorithms, we intend to use Theorem 1 only to guide us in the following way, so we suppress those details. Suppose we consider an optimization problem captured by function $f$ (e.g., Adwords, Knapsack, etc.); we will use the fact that an "online version" of $f$ (where the inputs arrive one at a time) often admits an efficient streaming or semi-streaming algorithm through the primal-dual framework (Mehta et al., 2007; Buchbinder et al., 2007; Buchbinder and Naor, 2009). If $f$ is symmetric, then the representation theorem above implies that $f$ can be computed by a pair of functions $S$ and $R$ in the sketching model. This reduces the problem of learning a uniform algorithm for $f$ to the problem of learning uniform algorithms $S$ and $R$. We invoke the symmetry of $f$ once again to conclude that $R$ must be a symmetric function as well — this implies that $R$ can be computed given only the set of values of the sketch function $S$ on the given input sequence. In particular, if the range of $S$ is a set of discrete values of size $k$, we only need to learn a function $R$ that can be computed from the $k$-bucket *histogram* of the values of $S$. If $S$ is computed in one-hot fashion, the count for each of the $k$ buckets in the histogram of values of $S$ is simply a sum, an eminently differentiable function!

