# OpenReview forum: "A new dog learns old tricks:  RL finds classic optimization algorithms"
_ICLR.cc/2019/Conference_

### Official Review · AnonReviewer2 · 2018-10-22
**Learning RL policies for Online Primal-Dual Algorithms**

**Rating:** 7
**Confidence:** 5

**Review:**

This paper studies the problem of *learning* online combinatorial algorithms via Reinforcement Learning. In particular, the paper studies three different well-studied problems, namely AdWords/Online Matching, Online Knapsack, and Secretary Problem. The common thread to all three problems is that they are special cases of Online Packing problem and that there exist optimal algorithms with theoretical guarantees. Moreover, all these problems have an algorithm based on the unifying Primal-Dual framework (*). The paper runs extensive experiments and shows that the learned RL policies resemble the algorithms created in theory by comparing many properties of the returned policy. Overall I think this paper tackles a very interesting question, is well-written and has extensive experiments.

I will detail my technical comments and questions to authors. I would especially appreciate detailed answers to (1), (2), (3). (4) and (5) are more open-ended questions and/or beyond the scope of the current paper. It can be viewed as feedback for some future work.

(1) (*) The paper starts with the claim that one of the key insights to this work is the primal-dual framework. Yet this has not been exploited in the paper; at least I can't see where it is used! Can the authors give more details? For example, one way I could see this being used is as follows. In the AdWords problem, the evolution of the Dual variable is well understood (See [Devanur, Kleinberg, Jain '13]). One can experiment to see how the dual values change over the run of the algorithm for the learned policy and compare that with how it changes in the analysis. If they are similar, then that is yet another evidence that the two algorithms are similar.

(2) One key point to note is that all three algorithms include only the "primal" observations in the state. This strengthens this work since all these algorithms, in theory, are analyzed by realizing that the primal algorithm can be interpreted as an appropriate process on the evolution of the dual variables.  Thus it seems like the RL policy is actually learning the optimal way to set the dual variables in the online phase. Is this true? I guess the experiments above can indeed verify this. If this is true, it implies that the key message of this paper is that RL algorithms can be used to learn algorithms that can be analyzed via the primal-dual framework. Right now, the authors stop short of this by saying this work is inspired from it. It would be good to see this taken to completion.

(3) It seems like there has been some work on using RL to learn algorithms in combinatorial optimization (see [Dai et al., NIPS 2017]). Can the authors discuss both in the rebuttal and in the paper on how their work compares and differs from this work?

(4) I wonder if the authors experimented with the i.i.d. arrival process for Knapsacks and/or Online Matching/Adwords. It is known that the theoretical algorithms for both these problems do much better than the pessimistic adversarial arrival order. It will be interesting to see if the RL policies also find this. On a related note, did the authors try this on b-matching with b=1? The problem tends to get easier as b is large and/or when bid/budget ratio is small in Adwords. However even when b=1, in theory, we can get 1-1/e [KVV '90].

(5) Finally, I am curious if the authors tested this on problems that are not packing but covering and/or mixed packing and covering problems. Online Set Cover is a candidate example. The other direction is also to test Online Minimization problems. Note that Online Min-cost matching is significantly harder than Online maximum weight matching. Moreover, Online Min-cost matching does not have a primal-dual analysis (to the best of my knowledge). The latter helps because if the RL policy fails to learn the best-known algorithm, then it is further evidence that it is indeed learning through the primal-dual framework.

Some minor comments:

Styling is not consistent throughout the paper. For example, there are places with the header followed by a colon sometimes, a period other times and sometimes no punctuation. Please make these consistent throughout the paper.

The fonts on the figures are very small to read. It would be easy on the reader if the sizes were made larger.

---

> ### Author Response · Authors · 2018-11-08
> **Request clarification for the suggested experiments**
>
> Thank you for your review, and the nice suggestion overall to drill down more into the connection to Primal-Dual algorithms. At this point (before responding to all the comments), we would like to clarify the experiment you have suggested in point (1). As you know, in the PD framework, the primal choices and the dual updates are intrinsically linked. In particular, for the Adwords problem, different choices of dual updates will yield different primal algorithms: e.g.,
> a) If we take the dual function for the advertiser side to be dual(u) = 0 if advertiser u has budget remaining and 1 if it has spent its budget fully, then the corresponding primal choice obtained via complementary slackness conditions (i.e.,  argmax_u {bid_{uv} * (1 - dual(u)) } ) yields the Greedy Algorithm.
> b) If we take the advertiser-side duals to be 1/(1-1/e) *(1-exp(x_u - 1)), where x_u = spend_u / budget_u, then we get the optimal Primal-Dual algorithm (MSVV/BJN).
>
> In the RL setting the neural net (NN) only gives us the decision of which u to allocate the incoming v (the “primal” decision). Therefore the question of “Does the RL [explicitly] learn the algorithm via duals”, although very intriguing, seems hard to pin down. We can think of a couple of interpretations of your question (i.e., what experiment to run):
>
> 1. Use the NN to make the primal allocation decision, but use the optimal PD updates ([BJN], in (b) above) to update the dual for the chosen advertiser. Then see if the dual for each advertiser unfolds, during a run on a specific instance (e.g., the upper triangular graph), similarly to how it would for MSVV/BJN. We’ll note that this really boils down to looking at the spend/budget curves for the advertisers over time during the run of the algorithm on the instance (since the BJN-dual is just a function of spend/budget).
>
> 2. A different interpretation is to assume that the RL uses some (hidden) dual variables to make its primal decisions, and to use the primal decisions it makes during its run on a specific instance to “reverse-engineer” the dual updates it uses. This seems a lot more difficult (we’re unsure if it can even be done reliably), and would involve something like:
>   a) At each time step the NN made a choice to give v to some u.
>   b) If we assume that the NN had some dual functions in mind --- dual_u(t)  or  maybe more explicitly dual_u(spend_u(t)/Budget_u) --- and made choices according to the complementary slackness condition argmax_u { bid_{uv} * (1-dual_u) }, then that gives a set of constraints on the functions "dual()".
>   c) Add all the constraints for all arriving vertices. and solve to find the best-fit dual() function.
>   d) Compare dual() to the BJN duals.
>
> Could you let us know if either of the above interpretations is what you had in mind (or had a different experiment in mind).
>
> [BJN] Buchbinder, Niv, Kamal Jain, and Joseph Seffi Naor. "Online primal-dual algorithms for maximizing ad-auctions revenue." European Symposium on Algorithms. Springer, Berlin, Heidelberg, 2007.
>
> [MSVV] Mehta, A., Saberi, A., Vazirani, U., & Vazirani, V. (2005, October). Adwords and generalized on-line matching. In 46th Annual IEEE Symposium on Foundations of Computer Science (FOCS'05) (pp. 264-273). IEEE.

---

> > ### Comment · AnonReviewer2 · 2018-11-09
> > **response**
> >
> > You are right that the primal and dual updates are very closely linked (which is why you should be able to see how the dual evolves corresponding to the primal actions chosen by your policy).
> >
> > The idea I had in mind was similar to (1). I think that will, to a certain extent, justify that the RL algorithm is indeed learning a primal-dual algorithm (as in the paper's claim). Essentially for many worst-case graphs (like the upper-triangular graph you suggest), we know how the dual variables evolve in theory. Your experiment in (1) will help visualize it for the learned policy.

---

> > > ### Author Response · Authors · 2018-11-16
> > > **response**
> > >
> > > (This is a continuation of the above response.)
> > >
> > > Point 4: Regarding the iid arrival question: We do look at iid distribution for Knapsack (most of the results for Knapsack, except those in 3.3.1 are in the iid setting, where values and sizes are picked iid). For Adwords, we have run experiments for the following settings, but we did not include them in this paper, due to space reasons and also because we felt they didn’t add more insights:
> > > - All bids are picked iid from U[0,1].
> > > - The graph is arbitrary, bids are 0/1, and the arrival order is random order. Then, by a result of [MPX], Balance achieves 1-o(1) ratio. Our learned agent also gets a ratio close to 1 (another evidence that it is actually doing Balance).
> > > - Degree-bounded graphs (from [NW]): We replicate the results from [NW] which show that Greedy is optimal for degree bounded graphs.
> > > - Power Law graphs.
> > >
> > > Excellent suggestion on trying for b=1 (bipartite matching). Will RL find KVV, given its connection to randomized primal dual [DJK]? We will try to pursue this in the future.
> > >
> > > Point 5: Nice suggestion to get more evidence towards the connection to Primal Dual. We will try to pursue this direction in the future.
> > > We have also fixed many of the typographic inconsistencies in the headings. Thank you for pointing these out.
> > >
> > > [DJK] Devanur, N., Jain, K., & Kleinberg, R. (2013). Randomized primal-dual analysis of ranking for online bipartite matching. ACM-SIAM symposium on Discrete algorithms.
> > >
> > > [MPX] Motwani, R., Panigrahy, R., & Xu, Y. (2006). Fractional matching via balls-and-bins. In Approximation, Randomization, and Combinatorial Optimization. Algorithms and Techniques (pp. 487-498). Springer, Berlin, Heidelberg.
> > >
> > > [NW] Naor, J. S., & Wajc, D. (2018). Near-optimum online ad allocation for targeted advertising. ACM Transactions on Economics and Computation (TEAC), 6(3-4), 16.

---

> > > > ### Author Response · Authors · 2018-11-16
> > > > **minor correction of citation in the previous comment**
> > > >
> > > > A minor correction:
> > > > In the above comment for point 4 (regarding the KVV algorithm), we meant to cite [DJK] (and not [BJN], or [BJK] which was a typo)
> > > >
> > > > [DJK] Devanur, N., Jain, K., & Kleinberg, R. (2013). Randomized primal-dual analysis of ranking for online bipartite matching. ACM-SIAM symposium on Discrete algorithms.

---

> > > > > ### Comment · AnonReviewer2 · 2018-11-25
> > > > > **Thanks!**
> > > > >
> > > > > Hi authors,
> > > > >
> > > > > Thanks for the clarifications and the additional experiments. We have seen them and will use them for our final evaluation.

---

> > > ### Author Response · Authors · 2018-11-16
> > > **response**
> > >
> > > Points 1 and 2 (which are connected):
> > > We believe that the key message you stated (“RL algorithms can be used to learn algorithms that can be analyzed via the primal-dual framework”) holds, but it seems hard to prove this theoretically, and we think new ideas may be needed before we can make that conclusion. We hope the current results will spur subsequent work towards establishing (or refuting) this formal connection. We agree that it would have been better for us to make this clearer: that we point out the potential connection, and find empirical evidence for three different problems which fall in the Primal-Dual framework, but we leave the question of proving or disproving it formally for future work. We have reworded the abstract to remove this confusion.
> > >
> > > We have run the experiments you suggested (thanks for the initial suggestion and the clarification). We have added the results in Appendix C (Fig 5, pg 14). Indeed, the “learned duals” (computed according to your suggestion) look quite similar to the duals from Balance! The duals are computed for runs on the “adversarial”/”upper-triangular” graph with 0/1 bids (Fig 4a), with 10 advertisers, each with a budget of 50 (and averaged over 100 runs). We have one plot per advertiser, showing how the duals unfold over time in the learned algorithm and in Balance. We present plots for 6 of the 10 advertisers (the ones who want the first 10%, 30%, 40%, 70%, 90&, and 100% of the ad slots that come through). One can see that the learned-duals are pretty close for the advertisers who desire a greater percentage of the ad slots, while somewhat off (but very similar in shape) for some of the advertisers who desire a small percentage of the ad slots.
> > >
> > > Point 3:
> > > Thanks for suggesting a closer look at the [Dai et al., NIPS 2017] reference. (To clarify, we did cite it in the related work section, but Google scholar citation had flipped the order of authors for some reason so may not have been apparent). It is a nice paper, and we have edited the paper to include additional discussion. There are some fundamental differences and we expand on them below.
> > >
> > > The [Dai et al., NIPS 2017] paper is focused is obtaining good heuristics for graph problems which possess some structure (for example, Erdos-Renyi graphs, Barabasi-Albert graphs, or realistic data, such as the Memetracker graph), i.e., where the instances are picked from a distribution. The goal is to beat known approximation algorithms for instances from the given distribution. The goal in this (our) paper is different; we aim to understand the types of algorithms that can be obtained via RL and to see whether the learned networks behave similarly to theoretically optimal algorithms. Problems such as adwords and online knapsack have succinctly described optimal algorithms and by training on the appropriate distributions, the learned agent is similar to the optimal algorithms.
> > > Another (maybe less fundamental) difference: [Dai et al.] tackles problems which are offline, i.e., the entire graph is known in advance, and the policy picks an incremental (“greedy”) solution by deciding which vertex to pick next. This is important for their solution since it uses an embedding technique for vertices based on their neighborhood structure. The problems we tackle are online, with the instance revealed incrementally over time and an item/vertex/allocation has to be made irrevocably upon arrival.

---

### Official Review · AnonReviewer1 · 2018-11-02
**A review on  "A new dog learns old tricks: RL finds classic optimization algorithms"**

**Rating:** 6
**Confidence:** 3

**Review:**

The overall goal of this paper is to solve online combinatorial optimization (OCO) problems using reinforcement learning. Importantly, the authors are not seeking to establish new results for unsolved problems, but instead they are motivated by analyzing and comparing the quality of solutions predicted by reinforcement learners with respect to the well-known near-optimal strategies for some OCO tasks. In doing so, the authors focused on an MDP framework using policy gradient and DQN methods. This framework was trained on three OCO tasks; online budget allocation, online knapsack, and the secretary problem. For each, the trained model is consistent with the near-optimal “handcrafted’’ algorithms.

The idea of checking whether a standard RL framework, without prior information about “how to” solve a given OCO task, is capable from experience to reach the performance of existing optimal strategies (especially primal-dual approaches), is clearly interesting. But I am not entirely convinced that the paper is making novel contributions in this direction. My comments are detailed below:

(1) OCO problems have been a subject of extensive research in online learning (see e.g. [1,2,3]). Notably, the main issues related to “input length independence” (Sec 1.1) and “adversarially chosen input distributions” (Sec 1.2) have already been addressed in online learning frameworks. Input length independence is related to “horizon-independence” in online learning (the number $T$ of trials is not known in advance), and well-known approaches have been developed for devising horizon-independent forecasters, or transforming a horizon-dependent forecaster into a horizon-independent one (see e.g. [4]). Also, the study of online learners with respect to different properties of input sequences (stochastic, adversarial, or hybrid), is a well-known topic of research which have been conceptualized and formalized with appropriate metrics (see e.g. [5,6]).

(2) Although the authors are interested in making connection between RL and “primal-dual approaches” in online learning, this connection was not made clear in the paper. Namely, the overall contribution was to show that deep RL architectures can compete with existing, handcrafted online strategies, on three specific tasks. But in order to make a concrete connection with primal-dual approaches, this study should be extended to more general primal (covering) and dual (packing) problems as described in [7], and the RL framework should be compared with “generic” online primal-dual algorithms also described in [7]. We may note in passing that the offline versions of the three tasks examined in the present paper belong to the approximation classes PTAS or APX. By contrast, the complexity class of general covering/packing problems is much higher (a constant approximation ratio is not achievable unless P=NP) and, to this point, it would be interesting to examine whether a standard deep RL framework can compete with existing online strategies (for example in [7,8]) on such hard problems.

(3) Even if we stick to the three problems examined in the paper, the neural nets vary between tasks, with different numbers of layers, different widths, different batch sizes, etc. On the one hand, it is legitimate to seek for an appropriate learning architecture for the input problem. On the other hand, such adjustments are conveying some prior knowledge about “how to” solve this problem using a deep RL model. Moreover, for knapsack and secretary tasks, additional knowledge about the history (i.e. state augmentation) is required for establishing convergence, but the resulting model is no longer a standard MDP. So, unless I missed the point about the overall goal of this paper, these different aspects are somewhat in contradiction with the idea of starting with a deep RL architecture with default settings, in which the varying components are essentially the states, transitions, and rewards of the MDP that encodes the problem description.

[1] Bubeck, S., Introduction to Online Optimization. Lecture Notes, Princeton University, 2011.

[2] Audibert, J-Y., Bubeck, S., and Lugosi, G. Minimax policies for combinatorial prediction games. In COLT, 2011.

[3] Rajkumar, A. and Agarwal, S. Online decision-making in general combinatorial spaces. In NIPS, 2014.

[4] N. Cesa-Bianchi and G. Lugosi. Prediction, Learning, and Games. Cambridge University Press, 2006.

[5] A. Rakhlin, K. Sridharan, and A. Tewari. Online learning: Stochastic, constrained, and smoothed adversaries. In NIPS, 2011.

[6] N. Buchbinder, S. Chen, J. Naor, O. Shamir: Unified Algorithms for Online Learning and Competitive Analysis. In COLT, 2012.

[7] N. Buchbinder and J. Naor. The Design of Competitive Online Algorithms via a Primal-Dual Approach. Foundations and Trends in Theoretical Computer Science, 2009.

[8] S. Arora, E. Hazan, and S. Kale. The Multiplicative Weights Update Method: a Meta-Algorithm and Applications, Theory of Computing, 2012.

---

> ### Author Response · Authors · 2018-11-09
> **Response to reviewer [2/2]**
>
> 2. On the connection to Primal-Dual:
> The reviewer made some excellent points regarding the primal-dual framework. We believe our work takes the first step to making a connection between RL and primal-dual algorithms. In future work, we hope to find deeper connections, as you have suggested here:
> a) Extending results to the entire packing-covering class of problems described in [BN, AHK],
> b) Finding stronger theoretical justifications for why RL learns Primal-Dual algorithms. (See also our comments to AnonReviewer2 on the same question).
> Your suggestion to see if the PTAS/APX nature of the offline problems we studied is important for the results to hold, is also interesting to pursue.
>
> 3. On the NN architecture:
> It is true that the neural nets differ between the tasks but we believe the differences are very minor. All our neural networks are standard feedforward neural networks with ReLU activation and are all trained via the REINFORCE algorithm. The differences in width, layers, etc. are just an artifact of working on the problems in parallel, and should be easily removable by using a common (largish) network. Of course, the input and output layers have to conform to the problem at hand (e.g., we get n bids and have to choose one of them in Adwords vs we get one item and have to decide to pick or not in knapsack).
>
> Regarding the comment on MDPs, we only do state augmentation when it is necessary. In the AdWords problem, we do not augment the state in any way. For the secretary problem, we only used the state augmentation in the “changing-iid” setting where each episode has its values drawn from a different distribution (we do not need augmentation for the fixed-iid, binary, and percentile settings). Recall that the optimal algorithm for the changing-iid (or adversarial setting) for secretary requires state augmentation, i.e. it knows the maximum value in the past (binary and percentile encode the past in its state; the fixed-iid does not require any past information). For the changing-iid distribution, the network would also need that information in some form to be competitive with the optimal algorithm.
> Similarly, for knapsack, the fixed-iid setting does not need past information, as the threshold can be learned in principle. But the changing-iid setting (Fig 7) does need the past information.
>
> Finally, we’ll also note that with or without state augmentation, our environment for secretary is not an MDP (since the reward really depends on the entire stream and past actions so it is not an MDP unless that information is encoded in the state somehow), so it is interesting in its own right that the RL can find close-to-optimal solutions very similar to the classic algorithms. In other words, the state augmentation was not done in order to obtain an MDP (as it is still not an MDP), but in order to make it feasible to learn the optimal algorithm. The Knapsack and Adwords environments are MDPs, (with and without augmentation) but only for the fixed-iid setting (since with changing distributions, the current state does not encode distribution over the item or vertex that arrives next, and therefore the distribution of the next state).
>
> [AHK] S. Arora, E. Hazan, and S. Kale. The Multiplicative Weights Update Method: a Meta-Algorithm and Applications, Theory of Computing, 2012.
>
> [BB] A. Blum and C. Burch. On-line learning and the metrical task system problem. In COLT, 1997.
>
> [BCNS] N. Buchbinder, S. Chen, J. Naor, O. Shamir: Unified Algorithms for Online Learning and Competitive Analysis. In COLT, 2012.
>
> [BE] Borodin, A., & El-Yaniv, R. (2005). Online computation and competitive analysis. Cambridge University Press.
>
> [BN] Buchbinder, N. and Naor, J.. The Design of Competitive Online Algorithms via a Primal-Dual Approach. Foundations and Trends in Theoretical Computer Science, 2009.
>
> [Bubeck] Bubeck, S., Introduction to Online Optimization. Lecture Notes, Princeton University, 2011.
>
> [RST] A. Rakhlin, K. Sridharan, and A. Tewari. Online learning: Stochastic, constrained, and smoothed adversaries. In NIPS, 2011.

---

> ### Author Response · Authors · 2018-11-09
> **Response to reviewer [1/2]**
>
> We would like to thank the reviewer for the detailed comments.
>
> 1. On the connection with Online Learning:
> Online Learning / Online Convex Optimization is, of course, a very well-studied field as the reviewer mentioned, including work on online combinatorial optimization (e.g., Ch. 6 in [Bubeck] as well as [2,3] from the referee review). However, we believe that the problem setting we study is not captured in that literature.
>
> In online convex (combinatorial) optimization, there is a new instance of a problem every time period (call it G(t)), and the algorithm has to pick a solution X(t) at every period, before actually knowing the parameters of the problem G(t). The goal is to come up with a policy that picks {X(t)} in such a way that there is low regret, on average, compared to a single solution X* used for all t. For e.g., (Ch. 6 in [Bubeck]), in bipartite matching, we get a new set of edge weights G(t) every time period, and we have to pick a matching X(t) before knowing G(t), so as to have no regret compared to a single matching M* used throughout. So, this kind of online convex optimization problem is a generalization of the classic “experts” or “Bandits” problem to the combinatorial setting.
> Our setting is different: There is one instance G (possibly picked from a distribution) which is revealed online (e.g., vertices of graph G arrive one at a time, or the candidates of G arriving one at a time), and the algorithm has to make a decision for each arriving element after seeing its parameters (e.g., where to match the t-th vertex, or whether to select the t-th candidate of not), and thus “build” a solution for the instance over time . The goal is to maximize the value of the solution for this instance compared to an optimal solution in hindsight. In other words, our setting is simply the classic Online Algorithms / competitive analysis setting (e.g. [BE]).
>
>
> As the reviewer rightly points out, there is work in the literature trying to connect the experts and the online competitive analysis settings. In particular ([BB,BCNS]) make a very nice connection between these two settings, for problems such as Metrical Task systems (MTS). Specifically, they show how to interpolate between the two settings. However, this connection still only works for certain types of online problems. In MTS, the feasible set is static throughout time. On the other hand, to encode the adwords problem in this framework, one would need to have feasible set which is updated to reflect which advertisers have exceeded their budget. (One alternative to this is to use the cost vector to reflect which advertisers have exceeded their budget but this requires an adversary which is not oblivious.) In summary, we do not think this connection to online learning fits our setting.
>
> Another point raised is that there is a lot of work on different properties of input sequences (stochastic, adversarial, or hybrid), e.g., [RST]. We agree that this is a well-studied topic in online learning and competitive analysis. If we interpreted the comment correctly (please let us know if not), then we believe there is a slight misunderstanding here. Here we are using “adversarially chosen input distributions” (Sec 1.2) in a very different way: We find universal (or high-entropy) distributions over inputs to train the RL. The distributions can be over input sequences which are (individually) stochastic as well (e.g., for Secretary problem). So, in our usage, we are the “adversary” trying to direct the RL agent to learn optimal behavior. We now see why this would be confusing (since it can be confused with the input order of individual sequences). We will try to change the wording to make this more clear.
>
>
> Regarding the point about input-length independence: we agree that online algorithms setting naturally provides input length independence, as does the online learning setting (e.g., work on horizon independence as you point out). Our goal here was indeed to learn optimal algorithms while sticking to input-length independent NNs. For example, we want to avoid using networks with different number of inputs as the length of the item-stream changes. Using the online competitive analysis framework helps in doing that (as mentioned above, we don’t think that the online learning setting is directly applicable). An additional point is that, as we mention in the appendix (Appendix F and G), this approach also extends to non-RL (e.g., supervised) learning, although indeed that connection is not part of our main results.

---

### Official Review · AnonReviewer3 · 2018-11-04
**Review for A new dog learns old tricks: RL finds classic optimization algorithms**

**Rating:** 6
**Confidence:** 3

**Review:**

This paper introduces a new framework to solve online combinatorial problems using reinforcement learning. The idea is to encode the current input, the global parameters, and a succinct data structure (to represent current states of the online problem) as MDP states. Such a problem can then be solved by deep RL methods. To train such models, the authors use a mixture of input distributions. Some come from hard distributions which are used to prove lower bounds in the TCS community, and the others are carefully constructed distributions to fool a specific set of algorithms. The authors made an important point that their algorithms are uniform in the TCS sense, i.e., the algorithm does not depend on the input length.
Furthermore, the authors show that the learned algorithms have similar behavior as those classical optimal algorithms that are first obtained in the online combinatorial optimization community.

The idea of using hard distributions to train the model is reasonable, but not extremely interesting/exciting to me since this is by now a standard idea in TCS. Moreover, in many cases (especially those problems that don't have a lower bound), it is even very hard to construct a hard distribution. In general, how should we use construct the input distribution in those cases? Can the proposed methods still generalize if we don't have an appropriate hard distribution? I would like to see some discussion/experiments along this line.

Moreover, it is unclear to me whether the methods proposed here can really learn *uniform* algorithms for the ADWORDS problem. To make the state length independent of the number of advertisers (n), the authors discretized the state space (see Appendix B). This approach might work for small (constant) n. But as we increase n to infinity, it seems to me that due to precision issues this approach will fail. If this is true, then in order to make this algorithm work, we also need to increase the precision of the real numbers used to describe the state as we increase n. If it is the case, why is the algorithm still uniform? If it is not the case, the authors need to provide extra experimental results to show that the effectiveness of the learned algorithm keeps unchanged, even if we keep increasing n and do not change the representation of the states.

It is an interesting observation that deep RL methods can actually learn an algorithm with similar behavior as optimal classical combinatorial optimization algorithms. However, there is no explanation for this, which is a little bit frustrating. Would this phenomenon be explained by the use of hard distributions? The paper can be strengthened by providing more discussions along this line.

Minor comments:

The caption of Table 2 seems to contradict its explanation (on top of page 14). Is the state space discretized or the number of advertisers changed?

---

> ### Author Response · Authors · 2018-11-09
> **Response to reviewer**
>
> We would like to thank the reviewer for the detailed comments.
>
> 1.On the “hard distributions”:
>
> a. Certainly, hard distributions are used in TCS to prove upper bounds on any algorithm’s performance. Here, our contribution is to make the connection that these distributions can also be used for training ML models. You mentioned that “the idea of using hard distributions to train the model is reasonable, but not extremely interesting/exciting to me since this is by now a standard idea in TCS.” Did we miss something? -- Could you please point us to some of the relevant literature?
> b. Our goal in this paper is to show that if one can find sufficiently hard input distributions for a problem then it is likely that a trained model will learn behaviours that are consistent with the optimal algorithm (for example, in the secretary problem and AdWords problem). The reviewer is correct in that hard distributions may not be known for all problems. For example, we do not know of a good lower bound distribution for the knapsack problem so for this reason, we had to handcraft our own.
> c. Given this, an interesting question is how to find hard distributions. This seems like a very nice question but also seems hard -- we are pursuing this direction in future work.
> d. We also want to remark that training the RL agent on specific (not hard) distributions have also led to some interesting consequences. For example: (i) by training the RL agent on the “fixed-i.i.d.” Secretary problem (where each value in a sequence is drawn from U(0,1) independently) it discovered the decreasing threshold algorithm (which is optimal); (ii) by training the RL agent on the i.i.d. knapsack problem (where each value and size are drawn i.i.d.) it learned the bang-per-puck algorithm, which is again optimal.
>
> 2. On “explanation for why RL discovers the classic algorithms”:
>
> It is indeed an interesting question. One hypothesis is that hard distributions capture the intrinsic difficulties of the problem and any algorithm which performs well on this distribution must exhibit a behaviour which is similar to the classic algorithms. In particular, we suspect that the space of optimal or near-optimal algorithms is very small and if the neural network converges to a good solution for the distribution then it must converge to an near-optimal algorithm. Our paper shows a number of examples for this: (1) in the adwords problem, the upper triangular graph is a hard distribution and any algorithm which achieves the optimal approximation ratio must be able to balance amongst the bidders and trade off spend against bid; (2) in the “fixed-i.i.d.” secretary problem, although the distribution itself may not be “hard”, any optimal algorithm must necessarily decrease its threshold over time. If there was a problem and a distribution for which there are multiple wildly different optimal solutions, then it would be interesting to see which solutions the neural network converges to (e.g., the one that is more succinct, or the one that is more generalizable to other distributions, etc.). We will take your suggestion to add some of this explanation in the paper.
>
> On this question, please also note our comments in the responses to the other two reviewers. The problems we study fall under the online Primal-Dual algorithms framework and we speculate that RL will be able to solve problems solvable in the primal-dual framework (which is also related to the multiplicative weights method). At this point, it's merely speculation, and proving this formally appears very hard. We believe that proving this would be a major theoretical result, and that we have made good initial steps empirically in this direction.
>
> 3. On “uniform algorithm for Adwords”:
> We will run additional experiments and address this comment soon. Thanks.

---

> > ### Author Response · Authors · 2018-11-20
> > **response on uniformity of AdWords**
> >
> > We have run experiments to verify whether or not RL can learn algorithms that are uniform for AdWords (without changing the representation of the states or increasing precision). In particular, we trained a model for the discretized state space only for inputs up to length 400 (i.e. 400 ad slots, with 20 advertisers each with a budget of 20). We used a discretization of k=5 for this model (i.e., the spent was discretized 5 ways so, e.g., all spends in [0, 0.2)*Budget are placed in the same bucket, etc.). Note that, due to this, one cannot get 100% of the Balance solution even for the training parameters -- the network gets 92% of the balance solution for the training parameters. If an RL learned algorithm is “uniform” then it should not degrade too far below 92% (compared to the Balance solution). In our experiments, we see that no matter how long the length of our input is, the quality of its solution never dropped to less than 84%, even as we scale up to 1 million ads.
> >
> > We have posted these experiment results below and added a corresponding table into the paper (Table 3 in the appendix). The semantics are always: (number of advertisers, budget per advertiser, number of ad slots) used in the experiment.
> > - (10, 10, 100) => 0.90 of Balance (i.e. for 10 bidders, each with a budget of 10 and 100 ad slots, our network achieves 0.90 of balance).
> > - (20, 20, 400) => 0.92 of Balance [* This was the set of training parameters]
> > - (30, 30, 900) => 0.88 of Balance
> > - (10, 2000, 20000) => 0.85 of Balance
> > - (10, 4000, 40000) => 0.85 of Balance
> > - (25, 4000, 100000) => 0.84 of Balance
> > - (50, 400, 20000) => 0.84 of Balance
> > - (100, 100, 10000) => 0.85 of Balance
> > - (100, 1000, 100000) => 0.85 of Balance
> > - (200, 100, 20000) => 0.85 of Balance
> > - (500, 50, 25000) => 0.85 of Balance
> > - (1000, 100, 100000) => 0.84 of Balance
> > - (25, 40K, 1M) => 0.84 of Balance.
> >
> > Note this observation that the learned algorithm is quite uniform (without changing the state representation or precision) is not entirely surprising. In the AdWords problem, for a fixed discretization there is an algorithm based on Balance that is oblivious to the number of advertisers and the number of ad slots and will achieve nearly the same approximation ratio (using a fixed, but fine discretization) as Balance without any discretization: Let b(i,j) be the 0-1 bid of advertiser i for ad slot j and x(i,j) be the fractional spend of advertiser i when ad slot j arrives. The discretized algorithm essentially replaces the operation argmin_i b(i,j) x(i,j) (or more generally, argmax_i b(i,j) psi(x(i,j)) for MSVV) with the following operation. First, rank each of the (fixed number of) grid points (b, x) according to its Balance/MSVV score above. When each ad slot is presented, it looks through all the grid points in this rank order to find the first grid point that has at least one advertiser and allocates the ad slot to one of those advertisers. It is fairly easy to see that the algorithm is still near-optimal modulo the discretization. (In fact, the proof in [MSVV] already discretizes the spend into “slabs”.)
> >
> > Hence, if the RL algorithm obtains an approximation to the discretized Balance algorithm above for short input lengths (which it seems to do) then it will continue to perform near-optimally for larger input lengths (i.e. number of advertisers and ad slots), as both the RL solution and the discretized Balance are oblivious to the number of advertisers or ad slots.
> >
> > [MSVV] Mehta, A., Saberi, A., Vazirani, U., & Vazirani, V. (2005, October). Adwords and generalized on-line matching. In 46th Annual IEEE Symposium on Foundations of Computer Science (FOCS'05) (pp. 264-273). IEEE.

---

> > > ### Comment · AnonReviewer3 · 2018-12-08
> > > **response to rebuttal**
> > >
> > > Thanks for the explanation and the extra experiments. Sorry for the confusion. By ``the idea of using hard distributions to train the model is reasonable, but not extremely interesting/exciting to me since this is by now a standard idea in TCS’’ I mean the idea of using hard distribution to prove lower bounds is a standard idea in TCS. Indeed it hasn’t appeared in training RL models before.
> > >
> > > I am satisfied with the explanation the authors provided for explaining why RL discovers classical algorithms and extra experiments performed for verifying the uniformity of the algorithms. Please do include these explanations and experimental results in later version of this paper.
> > >
> > > Overall, the rebuttal has mostly resolved my concerns, and thus I have changed my score to 6.

---

### Meta-Review · Area_Chair1 · 2018-12-13
**Not entirely novel but still very interesting approach**

**Confidence:** 4
**Recommendation:** Accept (Poster)

**Metareview:**

This paper is concerned with solving Online Combinatorial Optimization (OCO) problems using reinforcement learning (RL). There is a well-established traditional family of approaches to solving OCO problems, therefore the attempt itself to solve them with RL is very intriguing, as this provides insights about the capabilities of RL in a new but at the same time well understood class of problems.

The reviewers agree that this approach is not entirely new. While past similar efforts take away some of the novelty of this paper, the reviewers and AC believe that still the setting considered here contains novel and interesting elements.

All reviewers were unconvinced that this work can provide strong claims about using RL to learn any primal-dual algorithm. This takes away some of the paper’s impact, but thanks to discussion the authors managed to clarify some “hand-wavy” claims and toned-down the claims that were not convincing. Therefore, it was agreed that the new revision still provides some useful insight into the RL and primal-dual connection, even without a complete formal connection.